# Single-nucleus transcriptomics reveal the differentiation trajectories of periosteal skeletal/stem progenitor cells in bone regeneration

Simon Perrin[1], Maria Ethel[1], Vincent Bretegnier[1], Cassandre Goachet[1], Cécile-Aurore Wotawa[1], Marine Luka[2,3], Fanny Coulpier[1], Cécile Masson[4,5], Mickael Ménager[2,3], Céline Colnot[1]*

[1]Univ Paris Est Creteil, INSERM, IMRB, Creteil, France; [2]Paris Cité University, Imagine Institute, Laboratory of Inflammatory Responses and Transcriptomic Networks in Diseases, Atip-Avenir Team, INSERM UMR 1163, Paris, France; [3]Labtech Single-Cell@Imagine, Imagine Institute, INSERM UMR 1163, Paris, France; [4]Bioinformatics Core Facility, Institut Imagine-Structure Fédérative de Recherche Necker, INSERM U1163, Paris, France; [5]INSERM US24/CNRS UAR3633, Paris Cité University, Paris, France

*For correspondence: celine.colnot@inserm.fr

Competing interest: The authors declare that no competing interests exist.

## eLife Assessment

This **fundamental** study generated a single-cell atlas of mouse periosteal cells under both steady-state and fracture healing conditions to address the knowledge gap regarding cellular composition of the periosteum and their responses to injury. Based on **convincing** transcriptome analyses and experimental validation, the authors identified the injury induced fibrogenic cell (IIFC) as a characteristic cell type appearing in the bone regeneration process and proposed that the IIFC is a progenitor undergoing osteochondrogenic differentiation. This study will provide a significant publicly accessible dataset to reexamine the expression of the reported periosteal stem and progenitor cell markers.

**Abstract** Bone regeneration is mediated by skeletal stem/progenitor cells (SSPCs) that are mainly recruited from the periosteum after bone injury. The composition of the periosteum and the steps of SSPC activation and differentiation remain poorly understood. Here, we generated a single-nucleus atlas of the periosteum at steady state and of the fracture site during the early stages of bone repair (https://fracture-repair-atlas.cells.ucsc.edu). We identified periosteal SSPCs expressing stemness markers (*Pi16* and *Ly6a*/SCA1) and responding to fracture by adopting an injury-induced fibrogenic cell (IIFC) fate, prior to undergoing osteogenesis or chondrogenesis. We identified distinct gene cores associated with IIFCs and their engagement into osteogenesis and chondrogenesis involving Notch, Wnt, and the circadian clock signaling, respectively. Finally, we show that IIFCs are the main source of paracrine signals in the fracture environment, suggesting a crucial paracrine role of this transient IIFC population during fracture healing. Overall, our study provides a complete temporal topography of the early stages of fracture healing and the dynamic response of periosteal SSPCs to injury, redefining our knowledge of bone regeneration.

## Introduction

The skeleton provides structural support and protection for internal organs in the vertebrate body. Bones can fracture but regenerate themselves efficiently without scarring. Bone regeneration is mediated by the action of resident skeletal stem/progenitor cells (SSPCs) from periosteum and bone marrow, and SSPCs from skeletal muscles adjacent to the fracture site (*Duchamp de Lageneste et al., 2018*; *Jeffery et al., 2022*; *Julien et al., 2020*; *Julien et al., 2021*). SSPCs are activated during the inflammatory phase of healing and differentiate into osteoblasts and/or chondrocytes to repair bone via a combination of intramembranous and endochondral ossification. During intramembranous ossification, SSPCs differentiate directly into osteoblasts, while during endochondral ossification SSPCs first differentiate into chondrocytes to form an intermediate cartilage template subsequently replaced by bone.

The periosteum, an heterogeneous tissue located on the outer surface of bones, is a crucial source of SSPCs during bone healing (*Duchamp de Lageneste et al., 2018*; *Jeffery et al., 2022*; *Debnath et al., 2018*; *Matthews et al., 2021*; *Ortinau et al., 2019*). Periosteal SSPCs exhibit a high regenerative potential. They display both osteogenic and chondrogenic potentials after injury compared to bone marrow SSPCs that are mostly osteogenic and skeletal muscle SSPCs that are mostly chondrogenic (*Duchamp de Lageneste et al., 2018*; *Jeffery et al., 2022*; *Julien et al., 2020*; *Julien et al., 2021*; *Matsushita et al., 2020*). Periosteal SSPCs (pSSPCs) are still poorly characterized and their response to bone injury remains elusive. Recent advances in single-cell/-nucleus transcriptomic analyses provided new insights into stem cell population heterogeneity and regeneration processes in many organs (*De Micheli et al., 2020*; *Guerrero-Juarez et al., 2019*; *Storer et al., 2020*). However, few studies have investigated bone fracture healing at the single-cell level, and these studies have focused on cultured cells or late stages of bone repair (*Julien et al., 2020*; *Sivaraj et al., 2022*). Therefore, a complete dataset of periosteum and bone regeneration is lacking and is essential to decipher the mechanisms of pSSPC activation and regulation. Here, we created a single-nucleus atlas of the uninjured periosteum and its response to bone fracture. We generated single-nucleus RNAseq (snRNAseq) datasets from the uninjured periosteum and from the periosteum and hematoma/callus at days 3, 5, and 7 post-tibial fracture. Our analyses thoroughly describe the heterogeneity of the periosteum at steady state and the steps of pSSPC activation and differentiation after injury. We show that pSSPCs represent a single population localized in the fibrous layer of the periosteum. Periosteal SSPCs can provide osteoblasts and chondrocytes for bone repair by first generating a common injury-induced fibrogenic cell (IIFC) population that can then engage into osteogenesis and chondrogenesis. We identified the gene networks regulating pSSPC fate after injury and IIFCs as the main population producing paracrine factors mediating the initiation of bone healing.

## Results

### Heterogeneity of the periosteum at steady state

To investigate the heterogeneity of the periosteum at steady state, we performed snRNAseq of the periosteum of wild-type mice (*Figure 1A and B*). Single-nucleus transcriptomics was previously shown to provide results equivalent to single-cell transcriptomics, but with better cell type representation and reduced digestion-induced stress response (*Machado et al., 2021*; *Selewa et al., 2020*; *Ding et al., 2020*; *Wen et al., 2022*). After filtering, we obtained 1189 nuclei, corresponding to eight cell populations: SSPCs (expressing *Pi16*), fibroblasts (expressing *Pdgfra*), osteogenic cells (expressing *Runx2*), Schwann cells (expressing *Mpz*), pericytes/smooth muscle cells (SMCs, expressing *Tagln*), immune cells (expressing *Ptprc*), adipocytes (expressing *Lpl*), and endothelial cells (ECs, expressing *Pecam1*) (*Figure 1C and D*, *Figure 1—figure supplement 1A*). We performed in-depth analyses of the SSPC, fibroblast, and osteogenic cell populations. Subset analyses of clusters 0–5 identified five distinct SSPC/fibroblast populations expressing *Pdgfra* and *Prrx1*: $Pi16^+$ $Ly6a$ (SCA1) $^+$ cells (cluster 0), $Csmd1^+$ cells (cluster 1), $Hsd11b1^+$ cells (cluster 2), $Cldn1^+$ cells (cluster 3), and $Luzp2^+$ cells (cluster 4) (*Figure 2A and B*). Cluster 0 is the only cell cluster containing cells expressing *Pi16* and stemness markers including *Ly6a* (SCA1), *Dpp4*, and *Cd34* (*Figure 2C and D*). CytoTrace scoring identified $Pi16^+$ cells as the population in the most undifferentiated state (*Figure 2E*). We performed in vitro CFU assays with sorted $GFP^+SCA1^+$ and $GFP^+SCA1^-$ cells isolated from the periosteum of $Prrx1^{Cre}$; $R26^{mTmG}$ mice, as *Prrx1* labels all SSPCs contributing to the callus formation including $Pi16^+$

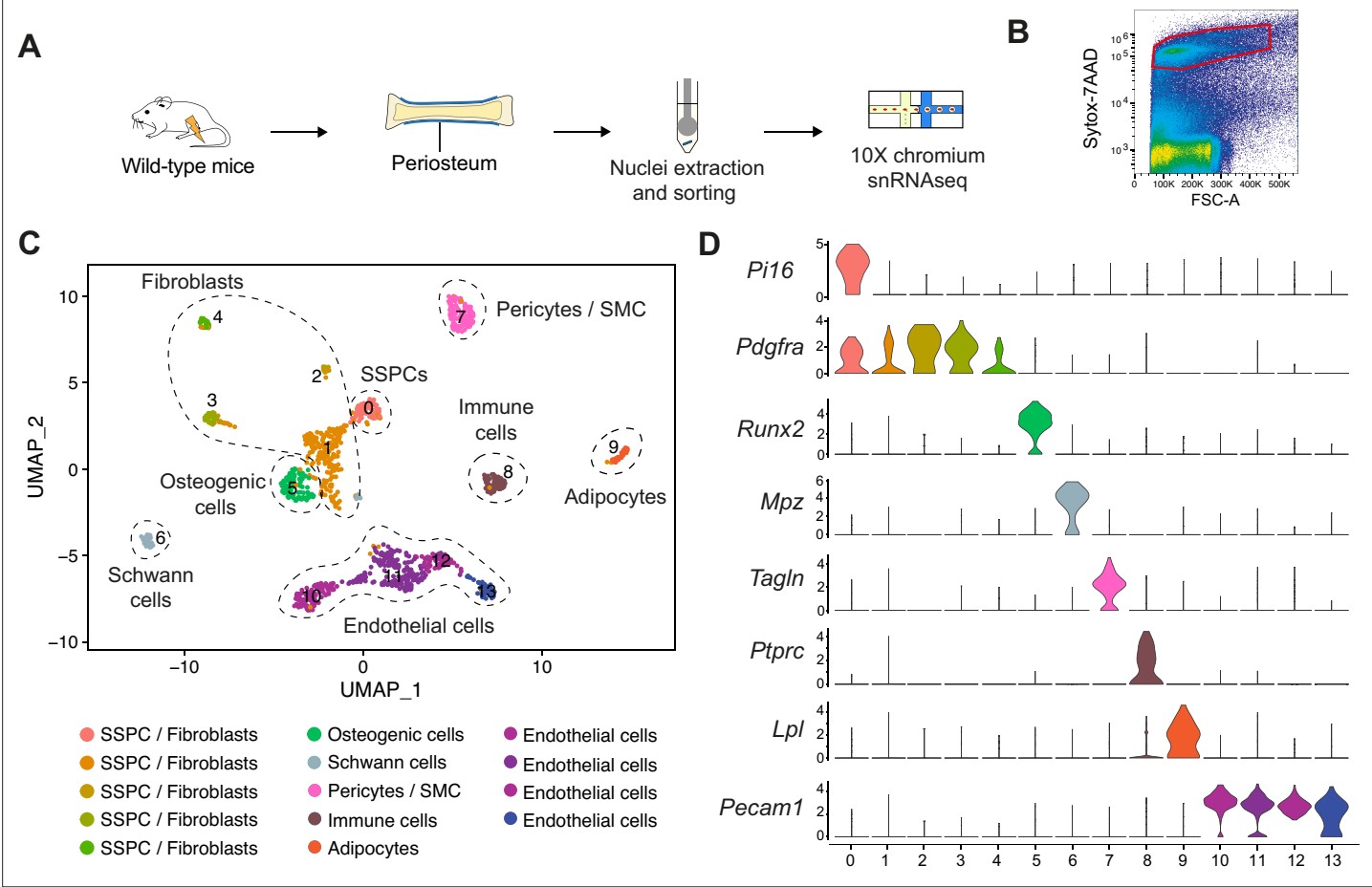

**Figure 1.** Heterogeneity of the periosteum at steady state. (**A**) Experimental design. Nuclei were extracted from the periosteum of uninjured tibia and processed for single-nucleus RNAseq. (**B**) Sorting strategy of nuclei stained with Sytox-7AAD for snRNAseq. Sorted nuclei are delimited by a red box. (**C**) UMAP of color-coded clustering of the uninjured periosteum dataset. Eight populations are identified and delimited by black dashed lines. (**D**) Violin plots of key marker genes of the different cell populations.

The online version of this article includes the following figure supplement(s) for figure 1:

**Figure supplement 1.** Dot plot of marker genes of the populations from uninjured periosteum.

cells (*Figure 2D*; *Duchamp de Lageneste et al., 2018*). Prrx1-GFP⁺ SCA1⁺ cells showed higher CFU potential compared to GFP⁺SCA1⁻ cells, confirming their stem/progenitor property (*Figure 2F and G*). Then, we grafted Prrx1-GFP⁺ SCA1⁺ and Prrx1-GFP⁺ SCA1⁻ periosteal cells at the fracture site of wild-type mice. Only GFP⁺SCA1⁺ cells formed cartilage after fracture indicating that GFP⁺SCA1⁺ cells encompass periosteal SSPCs with osteochondrogenic potential (*Figure 2H*). We explored the expression of other known markers of periosteal SSPCs, including *Ctsk*, *Acta2* (αSMA), *Gli1*, and *Mx1*, but no marker was fully specific to one cell cluster (*Figure 2—figure supplement 1*; *Duchamp de Lageneste et al., 2018*; *Jeffery et al., 2022*; *Debnath et al., 2018*; *Matthews et al., 2021*; *Ortinau et al., 2019*; *Tournaire et al., 2020*; *Gao et al., 2019*; *Chan et al., 2015*; *He et al., 2017*; *Böhm et al., 2019*).

## The fracture repair atlas

To investigate the periosteal response to bone fracture, we collected injured periosteum with hematoma /callus at days 3, 5, and 7 post-fracture, extracted the nuclei and processed them for snRNAseq (*Figure 3A*). We combined the datasets with the uninjured periosteum from *Figure 1* and obtained a total of 6213 nuclei after filtering. The combined dataset was composed of 25 clusters corresponding to 11 cell populations: SSPCs (expressing *Pi16*), IIFCs (expressing ECM-related genes including *Postn*), osteoblasts (expressing *Ibsp*), chondrocytes (expressing *Col2a1*), osteoclasts (expressing *Ctsk*), immune cells (expressing *Ptprc*), Schwann cells (expressing *Mpz*), endothelial cells (expressing

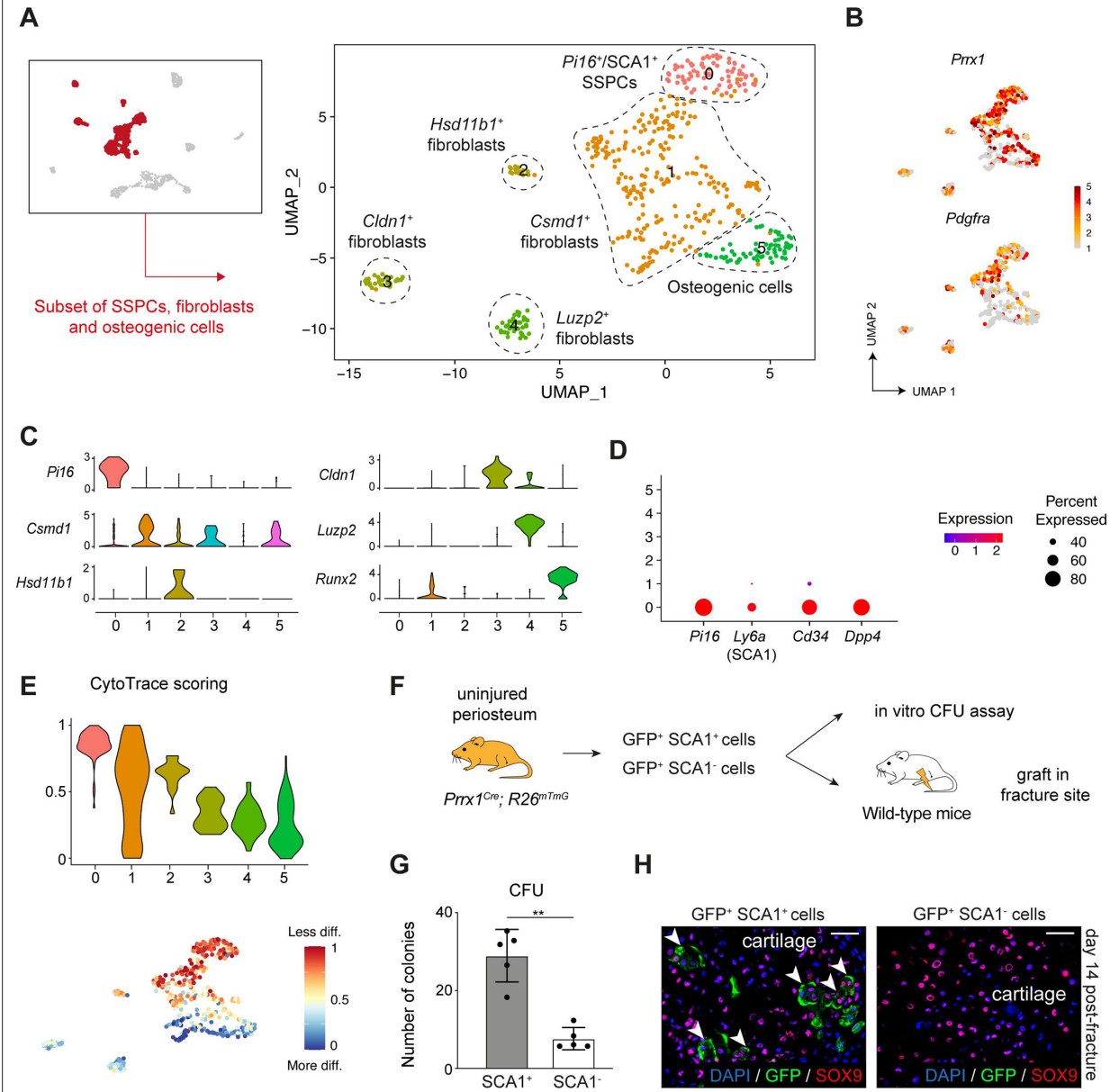

**Figure 2.** Identification of periosteal skeletal stem/progenitor cells in the intact periosteum. (**A**) UMAP of color-coded clustering of the subset of SSPCs/fibroblasts. (**B**) Feature plots of *Prrx1* and *Pdgfra* in the subset of SSPCs/fibroblasts. (**C**) Feature plots of key marker genes of the different cell populations. (**D**) Dot plot of the stemness markers *Pi16, Ly6a* (SCA1), *Cd34,* and *Dpp4.* (**E**) Violin and feature plots of CytoTrace scoring in the subset of SSPCs/fibroblasts, showing that SCA1 expressing SSPCs (cluster 0) are the less differentiated cells in the dataset. (**F**) Experimental design: GFP⁺ SCA1⁺ and GFP⁺ SCA1⁻ were isolated from uninjured tibia of *Prrx1^{Cre}; R26^{mTmG}* mice and used for in vitro CFU assays or grafted at the fracture site of wild-type mice. (**G**) In vitro CFU assay of murine periosteal Prrx1-GFP⁺ SCA1⁺ and Prrx1-GFP⁺ SCA1⁻ cells (n = 5 biological replicates from 2 distinct experiments). (**H**) High magnification of SOX9 immunofluorescence of callus section 14 days post-fracture showing that GFP⁺SCA1⁺ cells contribute to the callus (white arrowheads) while GFP⁺ SCA1⁻ cells are not contributing (n = 3 per group).

The online version of this article includes the following figure supplement(s) for figure 2:

**Figure supplement 1.** Expression of known skeletal stem/progenitor cell (SSPC) markers in the periosteum at steady state.

*Pecam1*), pericytes (expressing *Rgs5*), SMCs (expressing *Tagln*), and adipocytes (expressing *Lpl*) (***Figure 3B and C***, ***Figure 3—figure supplements 1 and 2***). Next, we observed the dynamics of the cell populations in response to bone fracture (***Figure 3D***, ***Figure 3—figure supplement 1B***). After injury, the percentage of SSPCs was strongly decreased and the percentage of IIFCs progressively increased (***Figure 3D and E***). The percentage of chondrocytes and osteoblasts increased from day

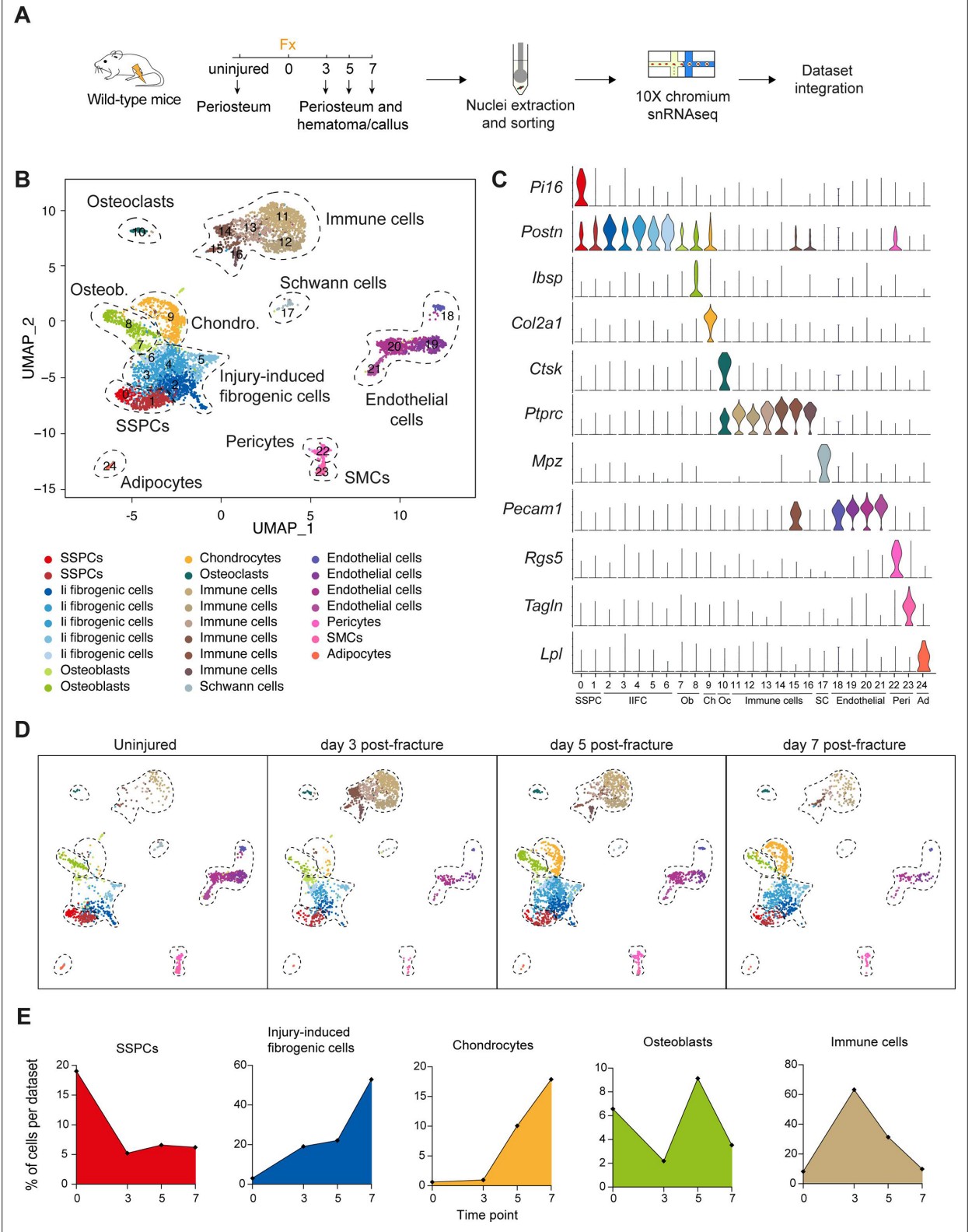

**Figure 3.** Periosteal response to fracture at single-nucleus resolution. (**A**) Experimental design. Nuclei were extracted from the periosteum of uninjured tibia and from the injured periosteum and hematoma/callus at days 3, 5, and 7 post-tibial fracture of wild-type mice and processed for single-nucleus RNAseq. (**B**) UMAP of color-coded clustering of the integration of uninjured, day 3, 5, and 7 datasets. Eleven populations are identified and delimited by black dashed lines. (**C**) Violin plots of key marker genes of the different cell populations. (**D**) UMAP of the combined dataset separated by time

*Figure 3 continued on next page*

*Figure 3 continued*

point. (**E**) Percentage of cells in skeletal stem/progenitor cell (SSPC), injury-induced fibrogenic cell, osteoblast, chondrocyte, and immune cell clusters in uninjured, day 3, 5, and 7 datasets.

The online version of this article includes the following figure supplement(s) for figure 3:

**Figure supplement 1.** Heterogeneity and dynamics of the cell populations in the fracture environment.

**Figure supplement 2.** Dot plot of marker genes of the populations from the combined fracture dataset.

3 post-fracture. Immune cells were drastically increased at day 3 after injury, before progressively decreasing at days 5 and 7 post-fracture (*Figure 3E*).

## Spatial organization of the fracture callus

To evaluate the spatial distribution of the main cell populations identified in the snRNAseq datasets, we performed in situ immunofluorescence and RNAscope experiments on uninjured periosteum and days 3, 5, and 7 post-fracture hematoma/callus tissues. In the uninjured periosteum, we detected *Pi16*-expressing SSPCs, *Postn*-expressing cells, OSX$^+$ osteoblasts, PECAM1$^+$ endothelial cells, and CD45$^+$ immune cells (*Figure 4A and B*). *Pi16*-expressing SSPCs were located within the fibrous layer, while *Postn*-expressing cells were found in the cambium layer and corresponded to *Runx2*-expressing osteogenic cells (*Figure 4—figure supplement 1A–C*). Although *Postn* expression was weak in uninjured periosteum, *Postn* expression was strongly increased in response to fracture, specifically in IIFCs (*Figure 4—figure supplement 1D and E*). At day 3 post-fracture, we observed periosteal thickening and the formation of a fibrous hematoma (*Figure 4C*). We did not detect *Pi16*-expressing SPPCs, consistent with the absence of cells expressing SSPC markers in the day 3 snRNAseq dataset compared to uninjured periosteum (*Figure 4—figure supplement 2*). POSTN$^+$ IIFCs and immune cells were the main populations present in hematoma and activated periosteum. Few IIFCs in the activated periosteum expressed SOX9 and OSX and only the periosteum was vascularized (*Figure 4D*). At days 5 and 7 post-fracture, the callus was formed mainly of fibrotic tissue, new bone formed on the periosteal surface at the periphery of the callus and small cartilage islets were detected in the center of the callus near the periosteal surface (*Figure 4E–H*). The fibrotic tissue contained mostly IIFCs, as well as immune and endothelial cells. We also observed SOX9$^+$ and OSX$^+$ cells in the fibrotic tissue, while the cartilage was solely composed of SOX9$^+$ chondrocytes. OSX$^+$ osteoblasts were the main cell population detected in the new bone at the periosteal surface. We also observed a progressive reduction of POSTN$^+$ cells and immune cells from days 5 to 7 and increased vascularization in the newly formed bone (*Figure 4E–H*).

## Periosteal SSPCs differentiate via an injury-induced fibrogenic stage

To understand the differentiation and fate of pSSPCs after fracture, we analyzed the subset of SSPC, IIFC, chondrocyte, and osteoblast clusters from the combined fracture dataset (*Figure 5A*, *Figure 5—figure supplement 1*). We performed pseudotime analyses to determine the differentiation trajectories, defining the starting point in the pSSPC population, as it corresponds to the uninjured and undifferentiated cells. We identified that pSSPCs differentiate in three stages starting from the pSSPC population (expressing *Ly6a*, *Pi16*, and *Cd34*), predominant in the uninjured dataset (*Figure 5B and C*). Periosteal SSPCs then transition through an injury-induced fibrogenic stage predominant at days 3 and 5 post-injury. In this intermediate fibrogenic stage, IIFCs express high levels of extracellular matrix genes, such as *Postn*, *Aspn,* and collagens. Subsequently, IIFCs differentiate into chondrocytes (expressing *Acan*, *Col2a1*, and *Sox9*) or osteoblasts (expressing *Sp7*, *Alpl*, and *Ibsp*), both predominant at days 5 and 7 (*Figure 5B and C*). We observed a parallel between pseudotime and the time points of the dataset, confirming that the differentiation trajectory follows the timing of cell differentiation (*Figure 5D*). These results show that pSSPCs respond to fracture via an injury-induced fibrogenic stage common to chondrogenesis and osteogenesis and independent of their final fate (*Figure 5E*). To visualize the transition of IIFCs toward chondrocytes and osteoblasts in the fracture callus, we performed co-immunofluorescence on day 5 post-fracture hematoma/callus. We observed a progressive increase in SOX9 and OSX signals in IIFCs at the fibrosis-to-cartilage and fibrosis-to-bone transition zones, respectively (*Figure 6A*). To functionally validate the steps of pSSPC activation, we isolated SCA1$^+$ GFP$^+$ pSSPCs from *Prrx1$^{Cre}$; R26$^{mTmG}$* mice, excluding endothelial cells (SCA1$^+$GFP$^-$) and

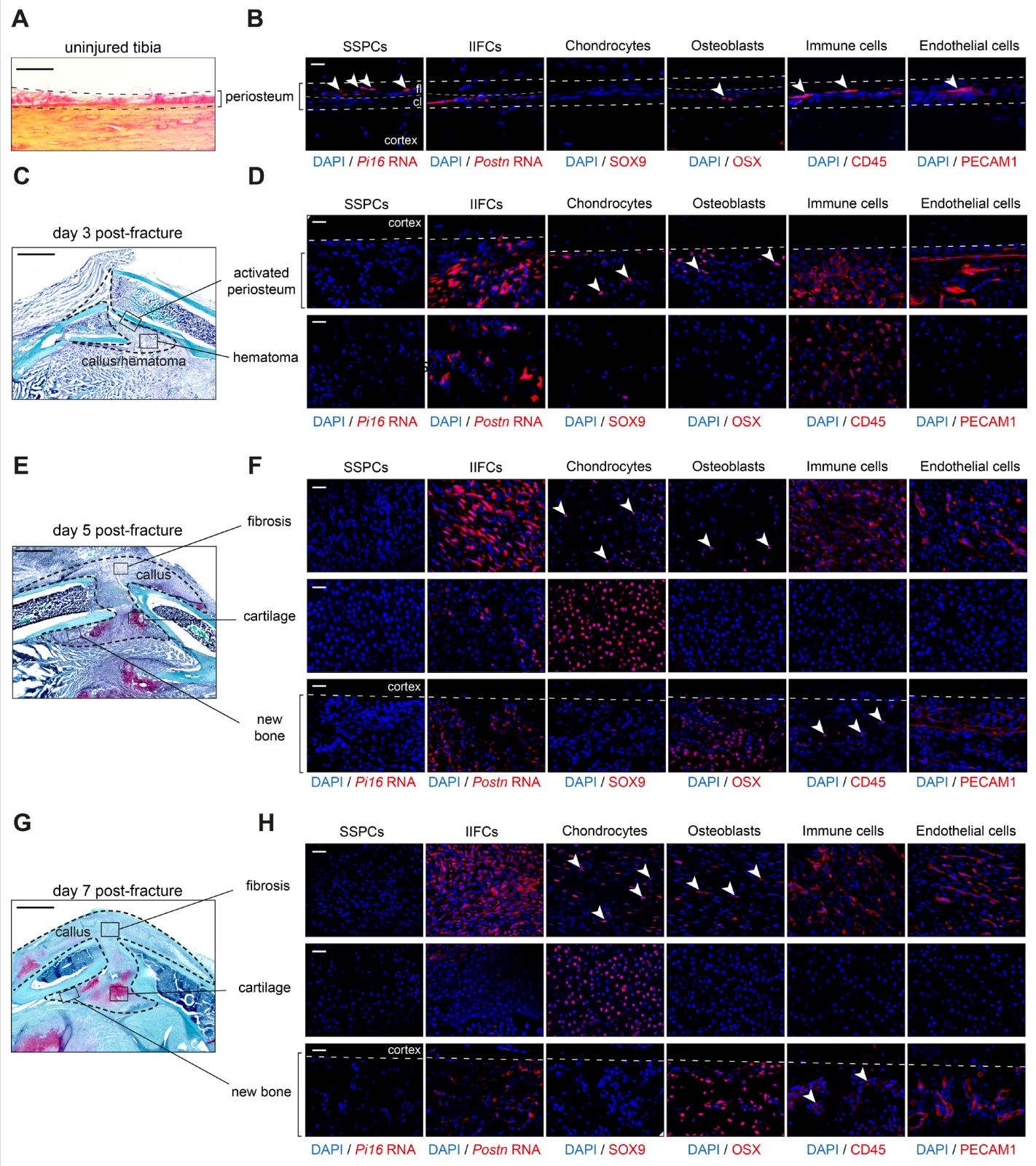

**Figure 4.** Cellular organization of the fracture callus. (**A**) Picrosirius staining of the uninjured periosteum. (**B**) Immunofluorescence and RNAscope on adjacent sections show the presence of SSPCs (*Pi16*-expressing cells) in the fibrous layer (fl), *Postn*-expressing cells in the cambium layer (cl), OSX+ osteoblasts, immune cells (CD45+), and endothelial cells (PECAM1+) in the periosteum (n = 3 per group). (**C**) Safranin'O staining of longitudinal callus sections at day 3 post-tibial fracture. (**D**) Immunofluorescence and RNAscope on adjacent sections show absence of skeletal stem/progenitor

*Figure 4 continued on next page*

*Figure 4 continued*

cells (SSPCs) (*Pi16*⁺), and presence of IIFCs (*Postn*⁺) and immune cells (CD45⁺) in the activated periosteum and hematoma at day 3 post-fracture. Chondrocytes (SOX9⁺, white arrowhead), osteoblasts (OSX⁺,white arrowhead), immune cells (CD45⁺), and endothelial cells (PECAM1⁺) are detected in the activated periosteum (n = 3 per group). (**E**) Safranin'O staining of longitudinal callus sections at day 5 post-tibial fracture. (**F**) Immunofluorescence and RNAscope on adjacent sections show injury-induced fibrogenic cells (IIFCs) (*Postn*⁺), chondrocytes (SOX9⁺, white arrowhead), osteoblasts (OSX⁺, white arrowhead), immune cells (CD45⁺), and endothelial cells (PECAM1⁺) in the fibrosis, chondrocytes (SOX9⁺) in the cartilage and osteoblasts (OSX⁺), immune cells (CD45⁺,white arrowhead), and endothelial cells (PECAM1⁺) in the new bone (n = 3 per group). (**G**) Safranin'O staining of longitudinal callus sections at day 7 post-tibial fracture. (**H**) Immunofluorescence and RNAscope on adjacent sections show IIFCs (*Postn*⁺), chondrocytes (SOX9⁺, white arrowhead), osteoblasts (OSX⁺, white arrowhead), immune cells (CD45⁺), and endothelial cells (PECAM1⁺) in the fibrosis, chondrocytes (SOX9⁺) in the cartilage and osteoblasts (OSX⁺), immune cells (CD45⁺, white arrowhead), and endothelial cells (PECAM1⁺) in the new bone (n = 3 per group). Scale bars: (**A–B–E**) 1 mm, (**B–D–F**) 100 μm.

The online version of this article includes the following figure supplement(s) for figure 4:

**Figure supplement 1.** Periosteal skeletal stem/progenitor cells do not express *Postn*.

**Figure supplement 2.** Absence of skeletal stem/progenitor cells in the injured periosteum.

pericytes (SCA1⁻GFP⁺), and grafted them at the fracture site of wild-type hosts (***Figure 6B***, ***Figure 6— figure supplement 1***). We observed that grafted GFP⁺ pSSPCs formed POSTN⁺ IIFCs at day 5 post-fracture (***Figure 6B***). Then, we isolated IIFCs, which correspond to GFP⁺ CD146⁻ cells from the day 3 post-fracture callus of *Prrx1^Cre^*; *R26^mTmG^* mice without contamination by pericytes (GFP⁺CD146⁺ cells) (***Figure 6C***, ***Figure 6—figure supplement 1***). We grafted the GFP+ IIFCs at the fracture site of wild-type hosts and showed that grafted cells formed bone and cartilage at day 14 post-fracture. These results confirmed that pSSPCs first become IIFCs that differentiate into osteoblasts and chondrocytes.

## Characterization of injury-induced fibrogenic cells

We performed in-depth analyses of the newly identified IIFC population. Gene Ontology (GO) analyses of upregulated genes in IIFCs (clusters 2–6) showed enrichment in GOs related to tissue development, extracellular matrix (ECM), and ossification (***Figure 7A***). We identified several ECM-related genes specifically upregulated in IIFCs, including collagens (*Col3a1*, *Col5a1*, *Col8a1*, *Col12a1*), *Postn*, and *Aspn* (***Figure 7B and C***, ***Figure 7—figure supplement 1***). We also identified GO terms related to cell signaling, migration, differentiation, and proliferation, classic hallmarks of injury response. Only a small subset of IIFCs undergo apoptosis, further supporting that IIFCs are maintained in the fracture environment giving rise to osteoblasts and chondrocytes (***Figure 7—figure supplement 2***). To further understand the mechanisms regulating SSPC activation and fate after injury, we performed gene regulatory network (GRN) analyses on the subset of SSPCs, IIFCs, osteoblasts, and chondrocytes using SCENIC package (Single Cell rEgulatory Network Inference and Clustering) (***Aibar et al., 2017***). We identified 280 activated regulons (transcription factor/TF and their target genes) in the subset dataset. We performed GRN-based tSNE clustering and identified SSPC, IIFC, chondrocyte, and osteoblast populations (***Figure 7D and E***, ***Figure 7—figure supplement 3A***). Fibroblasts from uninjured periosteum (*Hsd11b1*⁺, *Cldn1*⁺, and *Luzp2*⁺ cells and corresponding to cluster 10 of ***Figure 5B***) clustered separately from the other populations, suggesting the absence of their contribution to bone healing. Analysis of the number of activated regulons per cell indicated that SSPCs are the most stable cell population (higher number of activated regulons), while IIFCs are the less stable population, confirming their transient state (***Figure 7F***). We then investigated cell population-specific regulons. SSPCs showed activated regulons linked to stemness, including Hoxa10 (16g), Klf4 (346g), Pitx1 (10g), and Mta3 (228g) (***Leclerc et al., 2023***; ***Nemec et al., 2017***; ***Takahashi and Yamanaka, 2006***), and immune response, including Stat6 (30g), Fiz1 (72g), and Stat5b (18g) (***Chen et al., 2011***; ***Kollmann et al., 2021***; ***Tsuruyama et al., 2010***; ***Figure 7G***). Osteoblasts and chondrocytes display cell-specific activated regulons including Sp7 (18g) and Sox9 (77g), respectively. We identified 21 regulons that we named fibro-core and that were upregulated specifically in the IIFC population (***Figure 7H and I***, ***Supplementary file 2***). Several fibro-core regulons, such as Meis1 (1556g), Pbx1 (11g), Six1 (20g), and Pbx3 (188g), are known to be involved in cell differentiation during tissue development and repair (***Le Grand et al., 2012***; ***Linares et al., 2015***; ***Rottkamp et al., 2008***; ***Wang et al., 2023***). Reactome pathway analysis showed that the most significant terms linked to IIFC-specific TFs are related to Notch signaling (***Figure 7J***, ***Supplementary file 3***). We confirmed that Notch signaling is increased in the IIFC stage (***Figure 7J***), suggesting its involvement in the fibrogenic phase of bone repair.

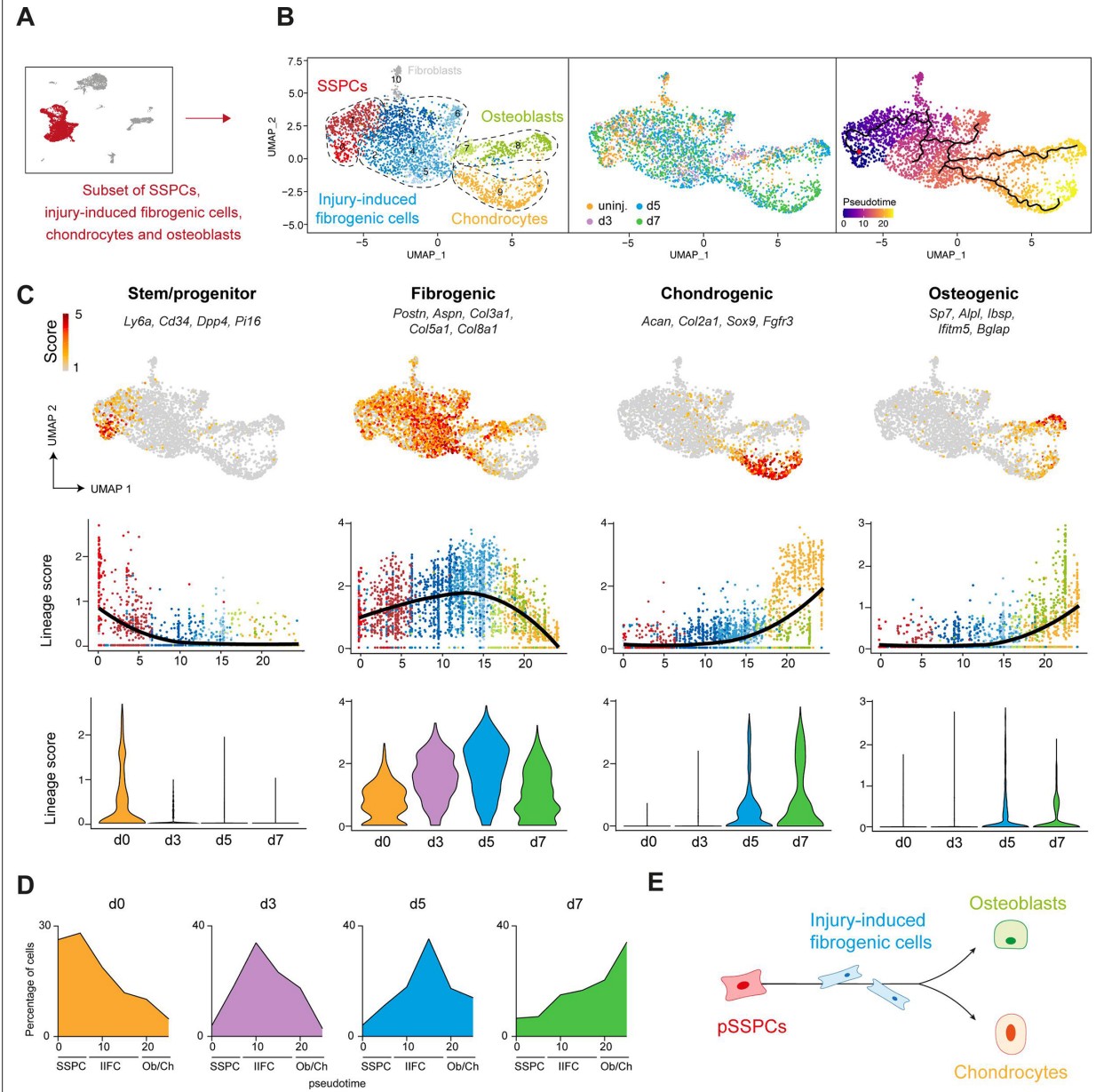

**Figure 5.** Periosteal skeletal stem/progenitor cells activate through a common fibrogenic state prior to undergoing osteogenesis or chondrogenesis. (**A**) SSPCs, injury-induced fibrogenic cells (IIFCs), chondrocytes, and osteoblasts from integrated uninjured, day 3, 5, and 7 post-fracture samples were extracted for a subset analysis. (**B**) UMAP of color-coded clustering (left), color-coded sampling (middle), and monocle pseudotime trajectory (right) of the subset dataset. The four populations are delimited by black dashed lines. (**C**) (Top) Feature plots of the stem/progenitor, fibrogenic, chondrogenic, and osteogenic lineage scores. (Middle) Scatter plot of the lineage scores along pseudotime. (Bottom) Violin plot of the lineage scores per time point. (**D**) Distribution of the cells along the pseudotime per time point. (**E**) Schematic representation of the differentiation trajectories of pSSPCs after fracture.

The online version of this article includes the following figure supplement(s) for figure 5:

**Figure supplement 1.** UMAP highlighting the distribution of periosteal fibroblasts in the combined fracture dataset.

## Distinct gene cores regulate the engagement of IIFCs in chondrogenesis and osteogenesis

We sought to identify the drivers of the transition of IIFCs to chondrocytes or osteoblasts. We identified two cores of regulons involved in chondrogenic differentiation. Chondro-core 1 is composed of 9 regulons specific to the transition of IIFCs to chondrocytes, including Maf (17g), Arntl (1198g), and Nfatc2 (37g), and chondro-core 2, composed of 14 regulons specific to differentiated chondrocytes

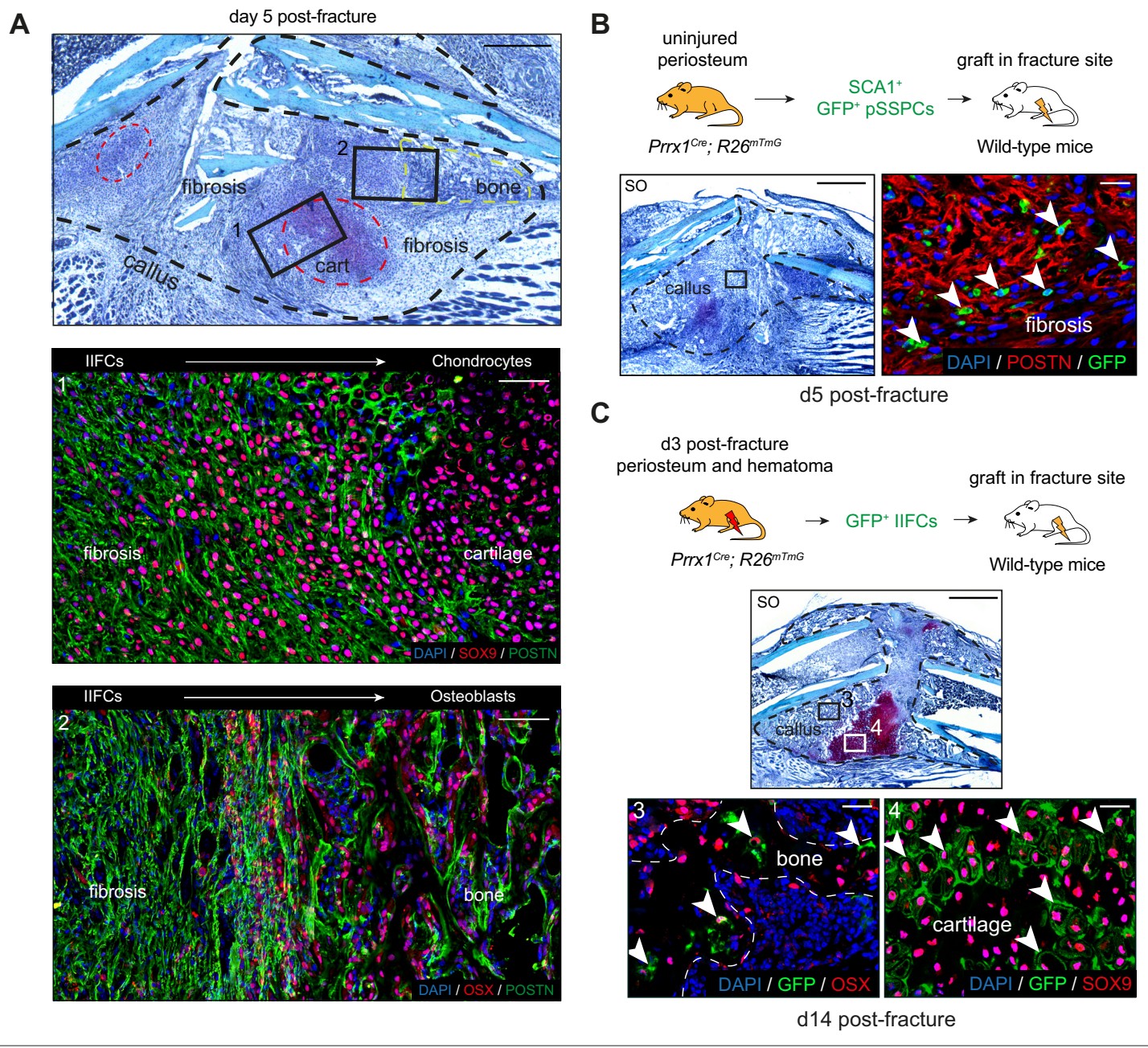

**Figure 6.** In vivo validation of periosteal skeletal stem/progenitor cell differentiation trajectories. (**A**) (Top) Representative Safranin'O staining on longitudinal sections of the hematoma/callus at day 5 post-fracture. The callus is composed of fibrosis, cartilage (red dashed line), and bone (green dashed line). (Middle, box 1) Immunofluorescence on adjacent section shows decreased expression of POSTN (green) and increased expression of SOX9 (red) in the fibrosis-to-cartilage transition zone. (Bottom, box 2) Immunofluorescence on adjacent section shows decreased expression of POSTN (green) and increased expression of OSX (red) in the fibrosis-to-bone transition zone (n = 3 per group). (**B**) Experimental design: GFP+ SCA1+ SSPCs were isolated from uninjured tibia of *Prrx1^Cre^; R26^mTmG^* mice and grafted at the fracture site of wild-type mice. Safranin'O staining of callus sections at day 5 post-fracture and high magnification of POSTN immunofluorescence of adjacent section showing that grafted GFP+ SSPCs contribute to the callus and differentiate into POSTN+ IIFCS (white arrowheads) (n = 4 per group). (**C**) Experimental design: GFP+ IIFCs from periosteum and hematoma at day 3 post-fracture tibia were isolated from *Prrx1^Cre^; R26^mTmG^* mice and grafted at the fracture site of wild-type mice. Safranin'O of callus sections at day 14 post-fracture and high magnification of OSX and SOX9 immunofluorescence of adjacent sections showing that grafted GFP+ injury-induced fibrogenic cells (IIFCs) contribute to the callus and differentiate into OSX+ osteoblasts (box 3, white arrowheads) and SOX9+ chondrocytes (box 4, white arrowheads) (n = 4 per group). Scale bars: low magnification: (**A**) 500 μm; (**B**, **C**) 1 mm. High magnification: 100 μm.

The online version of this article includes the following figure supplement(s) for figure 6:

**Figure supplement 1.** Validation of skeletal stem/progenitor cell (SSPC) and injury-induced fibrogenic cell (IIFC) sorting strategies.

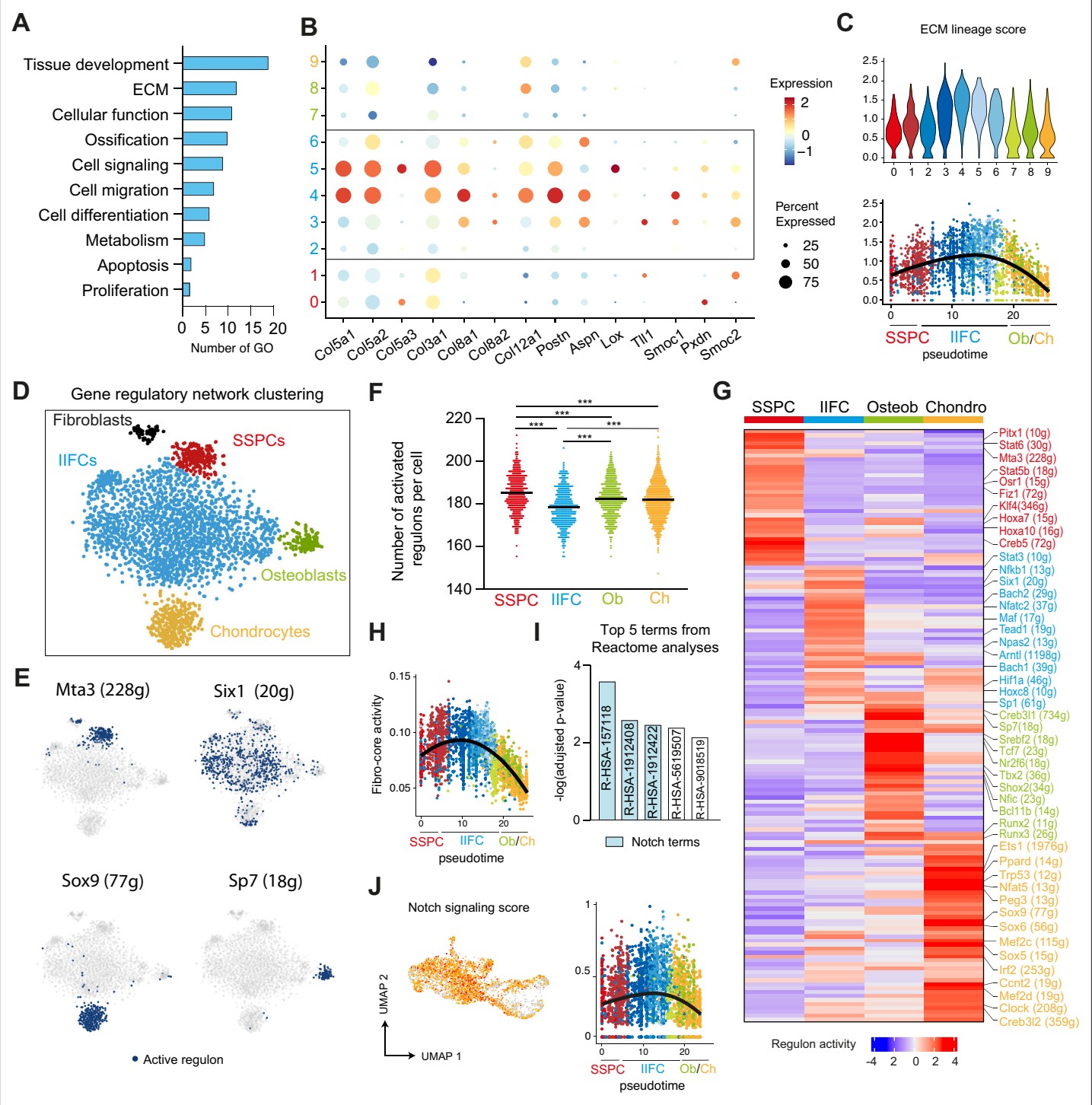

**Figure 7.** Characterization of injury-induced fibrogenic cells. (**A**) Gene Ontology analyses of upregulated genes in injury-induced fibrogenic cells (IIFCs) (clusters 2–6 of UMAP clustering from *Figure 5*). (**B**) Dot plot of extracellular matrix (ECM) genes in UMAP clustering from *Figure 5*. (**C**) Feature plot per cluster and scatter plot along pseudotime of the mean expression of ECM genes. (**D**) Gene regulatory network (GRN)-based tSNE clustering of the subset of skeletal stem/progenitor cells (SSPCs), IIFCs, chondrocytes, and osteoblasts. (**E**) Activation of Mta3, Six1, Sox9, and Sp7 regulons in SSPCs, IIFCs, chondrocytes, and osteoblasts. Blue dots mark cells with active regulon. (**F**) Number of regulons activated per cell in the SSPC, IIFC, osteoblast (Ob), and chondrocyte (Ch) populations. Statistical differences were calculated using one-way ANOVA. ***p-value<0.001. (**G**) Heatmap of activated regulons in SSPC, IIFC, osteoblast (osteob), and chondrocyte (chondro) populations. (**H**) Scatter plot of the activity of the combined fibrogenic regulons along monocle pseudotime from *Figure 5*. (**I**) Reactome pathway analyses of the fibrogenic regulons shows that the three most significant terms are related to Notch signaling (blue). (**J**) Feature plot in Seurat clustering and scatter plot along monocle pseudotime of the Notch signaling score.

*Figure 7 continued on next page*

*Figure 7 continued*

The online version of this article includes the following figure supplement(s) for figure 7:

**Figure supplement 1.** Injury-induced fibrogenic cells (IIFCs) are expressing extracellular matrix (ECM)-related genes.

**Figure supplement 2.** Injury-induced fibrogenic cells (IIFCs) do not undergo apoptosis.

**Figure supplement 3.** Activity of lineage specific regulons (Mta3, Six1, Sox9 and Sp7) in the UMAP Seurat clustering of SSPCs, IIFCs, chondrocytes and osteoblasts.

(*Figure 8A*, *Figure 8—figure supplement 1A*). Chondro-core 1 regulons are known to be regulators of the circadian clock (Npas2, Arntl) or of the T/B-cell receptor cellular response (Bach2, Nfact2, Nfkb1). STRING network analysis showed that TFs from the chondro-core 1 are interacting with each other and are at the center of the interactions between the chondro-core 2 TFs, including Sox9, Trp53, and Mef2c (*Figure 8B*, *Figure 8—figure supplement 1B*). We observed that chondro-core 1 is only transiently activated when IIFCs are engaging in chondrogenesis and precedes the activation of chondro-core 2 (*Figure 8C and D*). Chondro-core 1 activity was high in early differentiated chondrocytes (low *Acan* expression) and progressively reduced as chondrocytes underwent differentiation, while chondro-core 2 activity was gradually increased as chondrocytes differentiate (*Figure 8D*). This suggests that transient activation of the chondro-core 1 allows the transition of IIFCs in chondrocytes. Then, we investigated the osteogenic commitment of IIFCs. We identified eight regulons forming the osteo-core and activated in IIFCs transitioning to osteoblasts, such as Tcf7 (23g), Bcl11b (14g), and Tbx2 (36g) (*Figure 8E*, *Figure 8—figure supplement 1C*). STRING network analysis showed that the genes with the strongest interaction with the osteo-core TFs are mostly related to Wnt signaling (*Figure 8F*). This reveals the role of Wnt signaling in this transition from early fibrogenic activation of pSSPCs to osteogenic differentiation during bone repair. We calculated the osteo-core activity and observed that it is gradually increased and maintained in osteogenic cells, showing that the osteo-core is required for the transition and maturation of IIFC into osteoblasts (*Figure 8G and H*). Overall, we identified distinct cores of regulons with distinct dynamics driving the transition of IIFCs into chondrocytes and osteoblasts.

## IIFCs mediate paracrine interactions during bone repair

Paracrine cell interactions are crucial drivers of tissue regeneration and stem cell activation. To identify key cell interactions during bone repair, we performed cell interaction analyses using CellChat package (*Jin et al., 2021*). We observed that IIFCs are one of the predominant sources of outgoing signals during bone repair and are also important receivers of signals, suggesting their central role in mediating cell interactions after fracture (*Figure 9A and B*). Endothelial cells were mostly receiving signaling, while chondrocytes, osteoblasts, and most immune cells exhibited reduced interactions with the other cell types in the fracture environment. IIFCs interact with all cell populations in the fracture environment, but the strongest interactions were with SSPCs and IIFCs (*Figure 9—figure supplement 1A*). CellChat analyses of the subset of SSPCs, IIFCs, osteoblasts, and chondrocytes confirmed that IIFCs are a major source and receiver population of paracrine signals after fracture (*Figure 9—figure supplement 1B*). We then analyzed the main secreted factors from IIFCs. IIFCs secreted periostin (*Postn*), BMPs (*Bmp5*), pleiotropin/PTN (*Ptn*), TGFβs (*Tgfb2, Tgfb3*), PDGFs (*Pdgfc, Pdgfd*), and angiopoietin-likes/ANGPTLs (*Angplt2, Angplt4*) (*Figure 9C and D*, *Figure 9—figure supplement 1C*). We observed differences in the dynamics of these factors as some of them peaked at day 3 post-fracture such as TGFβ, while others peak at day 5 such as BMP, POSTN, PTN, and ANGPLT (*Figure 9E*). We assessed the dynamics of ligand and receptor expression and observed that ligand expression was increased during the IIFC phase and specific to the fracture response (*Figure 9F*). Receptor expression was high in both SSPCs and IIFCs, and receptors were expressed from steady state, suggesting that SSPCs can receive signals from IIFCs. Analysis of SSPC incoming signals showed that IIFCs produce paracrine factors that can regulate SSPCs (*Figure 9G*), indicating that they contribute to SSPC recruitment after fracture.

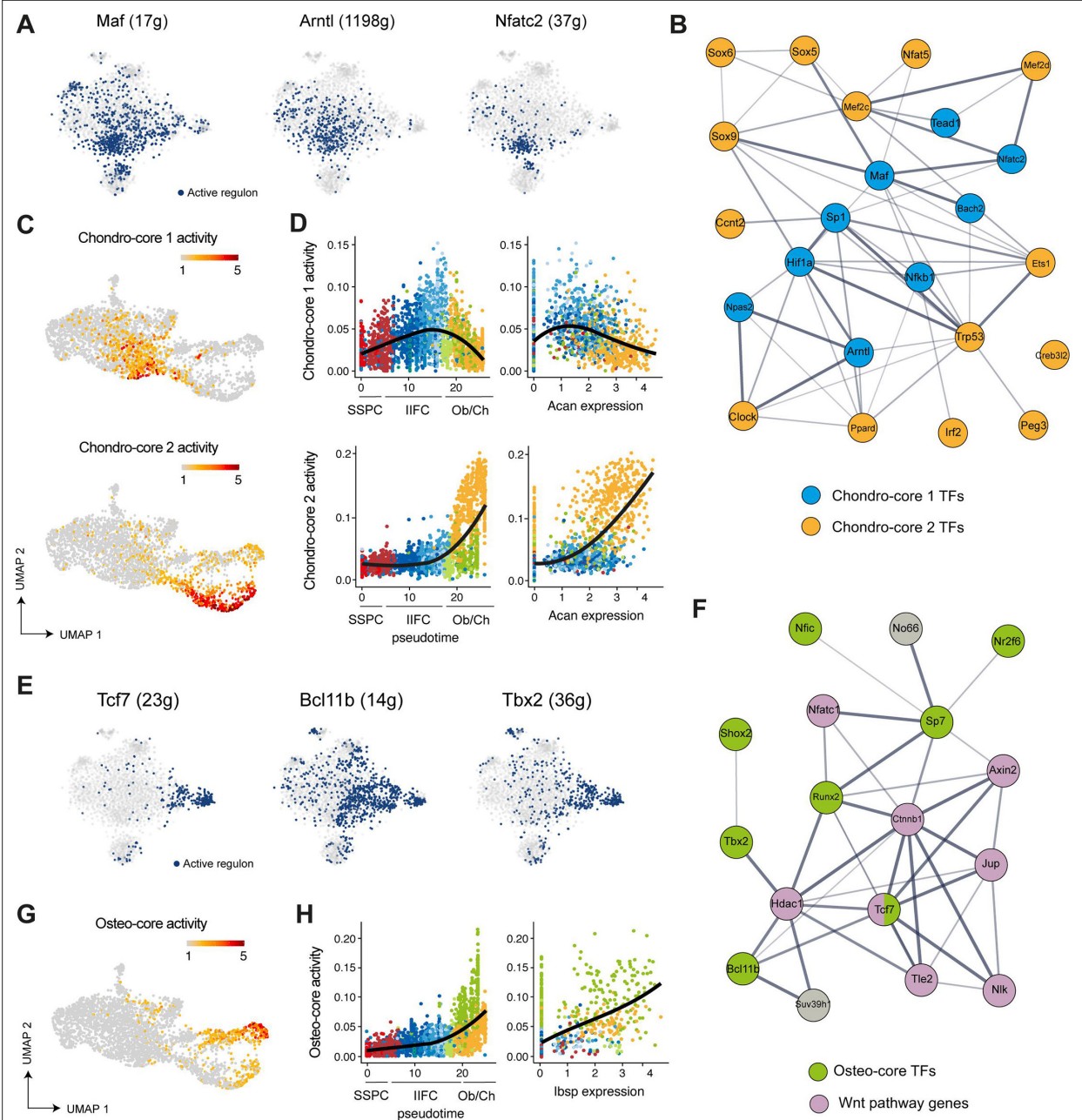

**Figure 8.** Gene regulatory network analyses identify gene cores driving fibrogenic to chondrogenic and osteogenic transitions. (**A**) Activation of Maf, Arntl, and Nfatc2 regulons in skeletal stem/progenitor cells (SSPCs), injury-induced fibrogenic cells (IIFCs), chondrocytes, and osteoblasts. (**B**) STRING interaction network of the chondro-core 1 and 2 transcription factors (blue and orange, respectively). (**C**) Feature plot of chondro-core 1 (top) and chondro-core 2 (bottom) activities in SSPCs, IIFCs, chondrocytes, and osteoblasts in Seurat UMAP from *Figure 5*. (**D**) Scatter plot of chondro-core 1 (top) and chondro-core 2 (bottom) activities along monocle pseudotime and *Acan* expression. (**E**) Activation of Tcf7, Bclb11b, and Tbx2 regulons in SSPCs, IIFCs, chondrocytes, and osteoblasts. (**F**) STRING interaction network of the osteo-core transcription factors (green) and their related genes shows that most of osteo-core related genes are involved in Wnt pathway (purple). (**G**) Feature plot of the osteo-core activity in SSPCs, IIFCs, chondrocytes, and osteoblasts in Seurat UMAP from *Figure 5*. (**H**) Scatter plot of osteo-core activity along monocle pseudotime and *Ibsp* expression.

The online version of this article includes the following figure supplement(s) for figure 8:

**Figure supplement 1.** Regulon activity in the subset of skeletal stem/progenitor cells (SSPCs), injury-induced fibrogenic cells (IIFCs), osteoblasts, and chondrocytes.

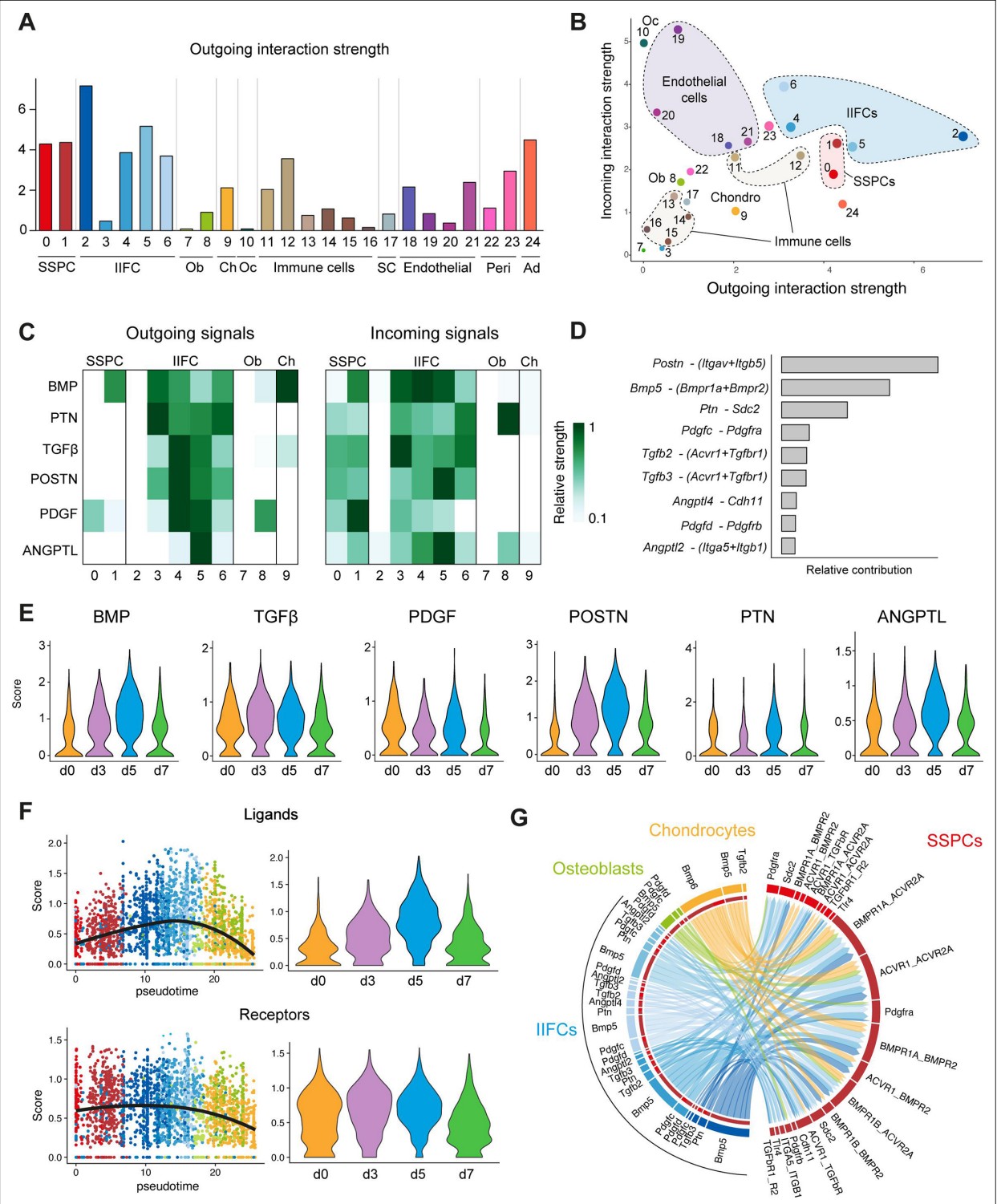

**Figure 9.** Injury-induced fibrogenic cells are the main source of paracrine factors after fracture. (**A**) Outgoing interaction strengths of the different cell populations of the fracture environment determined using CellChat package. (**B**) Comparison of incoming and outgoing interaction strengths across SSPC, IIFC, chondrogenic, and osteogenic populations. (**C**) Outgoing and incoming signaling from and to SSPCs, IIFCs, chondrocytes, and osteoblasts. (**D**) Cell–cell interactions identified between SSPCs, IIFCs, chondrocytes, and osteoblasts. (**E**) Violin plots of the score of BMP, TGFβ, PDGF, POSTN, PTN, and ANGPTL signaling per time point. (**F**) Scatter plot along pseudotime and violin plot per time point of the mean expression of the ligand and receptors involved in signaling from IIFCs. (**G**) Circle plot of the interactions between SSPCs, IIFCs, chondrocytes, and osteoblasts, showing that most signals received by SSPCs are coming for IIFCs. Ob: osteoblasts; Oc: osteoclasts; Ch: chondrocytes; SC: Schwann cells; Ad: adipocytes.

*Figure 9 continued on next page*

*Figure 9 continued*

The online version of this article includes the following figure supplement(s) for figure 9:

**Figure supplement 1.** Paracrine interactions from injury-induced fibrogenic cells (IIFCs).

## Discussion

The periosteum is the main driver of bone regeneration. Yet the periosteum composition and its response to bone fracture are poorly described. Here, we used single-nucleus transcriptomics to understand the heterogeneity of the periosteum at steady state and the changes occurring within the periosteum after bone fracture. We developed a protocol to extract nuclei from freshly dissected periosteum, allowing us to capture their intact transcriptomic profile without enzymatic digestion-induced stress (*Machado et al., 2021*; *Ding et al., 2020*; *van den Brink et al., 2017*). In addition, we performed snRNAseq without sorting specific populations, allowing the identification of all cell types located in the periosteum and the fracture environment from the early stages of repair. Our study provides the first complete fracture repair atlas, a key tool in understanding bone regeneration.

First, we described the heterogeneity of the periosteum at steady state. While previous studies performed scRNAseq on sorted periosteal cell populations, our dataset uncovers the diversity of cell populations within the periosteum. We describe fibroblast populations, as well as tissue-resident immune cells, adipocytes, and blood vessel/nerve-related cells. We identified one SSPC population of undifferentiated cells localized in the fibrous layer of the periosteum and expressing markers such as SCA1, *Dpp4,* and *Pi16,* known to mark fibroblasts with stemness potential (*Buechler et al., 2021*). No markers previously described such as *Ctsk* or *Gli1* were fully specific to one cell cluster and of the undifferentiated SSPC population, suggesting that these markers may label heterogeneous cell populations (*Jeffery et al., 2022*; *Debnath et al., 2018*; *Matthews et al., 2021*; *Chan et al., 2015*).

After fracture, the composition of the periosteum changes drastically, with the appearance of IIFC and immune cells from day 3 post-fracture, and of chondrocytes and osteoblasts from day 5 post-fracture. Previous studies based on in vitro analyses and in vivo lineage tracing demonstrated that the periosteum displays the unique potential to form both bone and cartilage after fracture (*Duchamp de Lageneste et al., 2018*; *Julien et al., 2020*; *Debnath et al., 2018*). Yet, it was still unknown if the periosteum contains a bipotent SSPC population or several SSPC populations with distinct potentials after injury. Here, we show that pSSPCs respond to injury and form bone and cartilage via a single trajectory emerging from SCA1$^+$ SSPCs (*Figure 10*). Periosteal SCA1$^+$ SSPCs become IIFCs, a state marked by a decreased expression of stemness markers and a strong expression of extracellular matrix genes, such as *Postn* and *Aspn* and activation of Notch-related TFs. Thus, bone injury does not induce an expansion of SCA1$^+$ SSPCs, but rather a transition toward IIFCs that progressively expand and represent the main cell population within the fracture callus until day 7 post-fracture. Following this fibrogenic step, IIFCs do not undergo cell death but either osteogenesis or chondrogenesis. This newly identified transient IIFC state represents the crossroad of bone regeneration and SSPC differentiation (*Figure 10*). In tissues such as skeletal muscle, an early transient fibrogenic response is also required for regeneration supporting muscle stem cell activation and differentiation (*Wosczyna et al., 2019*). During bone repair, this initial fibrogenic step is an integral part of the SSPC differentiation process, and a transitional step prior to osteogenesis and chondrogenesis.

GRN analyses identified TFs regulating SSPC response to fracture and involved in several cardinal signaling pathways including Notch and Wnt. Previous studies reported the role of Notch and Wnt in bone repair (*Matsushita et al., 2020*; *Lee et al., 2021*; *Wang et al., 2016*; *Dishowitz et al., 2012*; *Kraus et al., 2022*; *Matthews et al., 2014*; *Novak et al., 2020*; *Cao et al., 2017*; *Minear et al., 2010b*; *Minear et al., 2010a*; *Kang et al., 2007*; *Komatsu et al., 2010*). Notch inactivation at early stages of repair leads to bone non-union while Notch inactivation in chondrocytes and osteoblasts does not significantly affect healing, correlating with our data showing that Notch is crucial during the IIFC phase, prior to osteochondral commitment (*Wang et al., 2016*). Our results show that Wnt activation occurs after Notch activation and is specific to IIFCs engaging in osteogenesis, confirming its crucial role as a regulator of osteogenesis (*Matsushita et al., 2020*; *Minear et al., 2010b*; *Minear et al., 2010a*; *Kang et al., 2007*; *Komatsu et al., 2010*). In addition, we identified a chondro-core of nine regulons transiently activated when IIFCs engage in chondrogenesis. Among these regulons, several are involved in the autonomous circadian clock, including *Npas2* and *Arntl* (Bmal1). Bmal1 was

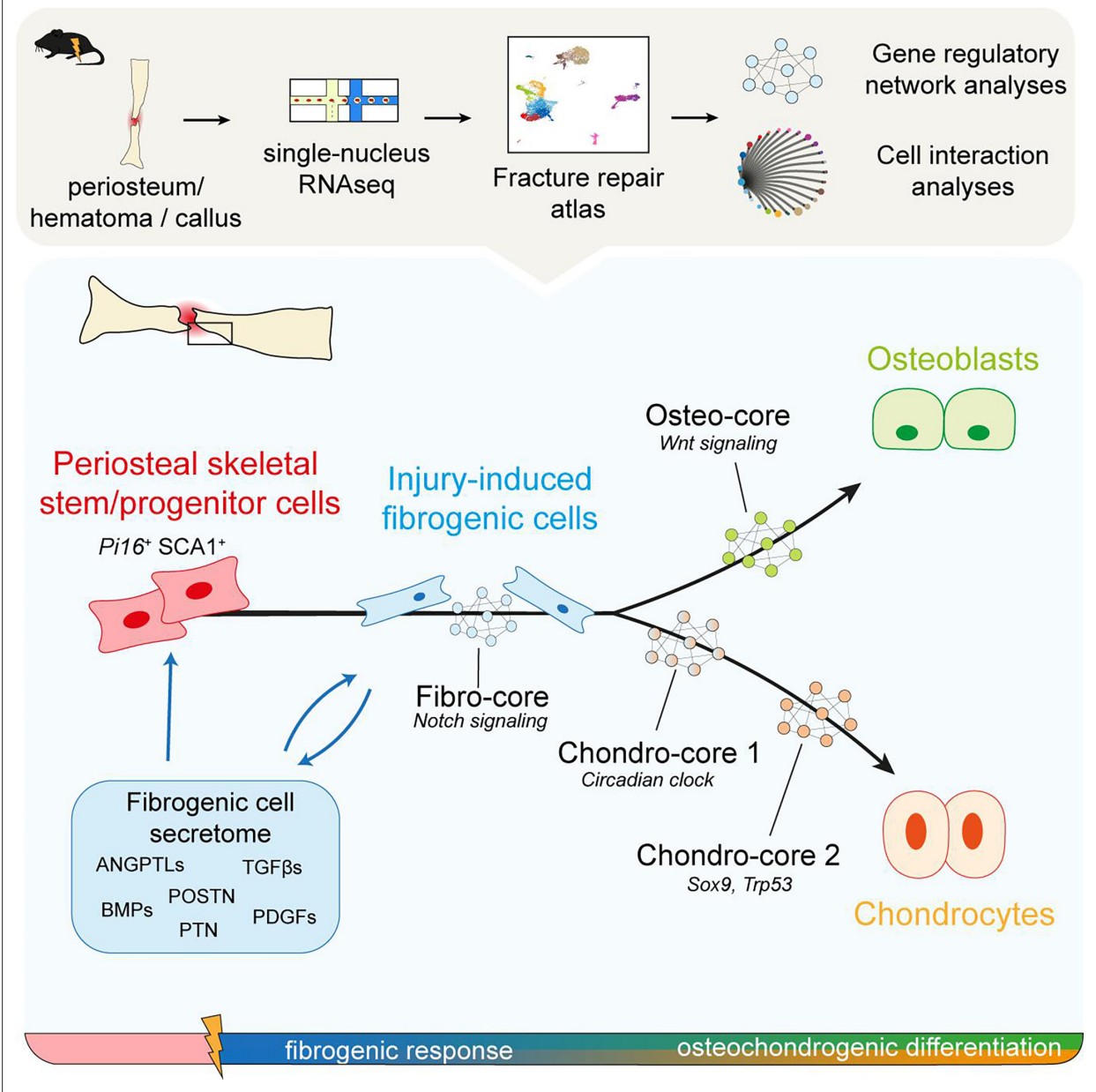

**Figure 10.** Activation and differentiation trajectories of periosteal skeletal stem/progenitor cells (SSPCs) after fracture.

previously shown to be a regulator of cartilage differentiation in bone development and homeostasis (*Chen et al., 2020*). Bmal1 inactivation leads to chondrocyte apoptosis and disruption of signaling involved in cartilage formation, including TGFβ, Ihh, HIF1α, and NFAT (*Dudek et al., 2023*; *Dudek et al., 2016*; *Ma et al., 2019*; *Takarada et al., 2012*). This suggests a role of the circadian clock genes as key regulators of chondrogenesis during bone repair. IIFCs are also revealed as crucial paracrine regulators of the fracture environment. IIFCs secrete factors including BMPs, PDGFs, TGFβs, and POSTN, known to be required for successful bone healing (*Duchamp de Lageneste et al., 2018*; *Julien et al., 2020*; *Gao et al., 2019*; *Xia et al., 2020*). These signals potentially regulate the interactions between IIFCs and all other cell types in the fracture environment, but primarily SSPCs and IIFCs themselves. Thus, IIFCs appear to contribute to the recruitment of SSPCs and promote their own maturation and differentiation.

Overall, our study provides a complete dataset of the early steps of bone regeneration. The newly identified SSPC activation pattern involves a coordinated temporal dynamic of cell phenotypes and

signaling pathways after injury. IIFCs emerge as a transient cell population that plays essential roles in the initial steps of bone repair. A deeper understanding of IIFC regulation will be crucial as they represent the ideal target to enhance bone healing and potentially treat bone repair defects.

## Materials and methods

### Key resources table

| Reagent type (species) or resource | Designation | Source or reference | Identifiers | Additional information |
|---|---|---|---|---|
| Strain, strain background (mice) | C57BL/6ScNj | Janvier Labs | | |
| Strain, strain background (mice) | B6.Cg-Tg(Prrx1-cre)1Cjt/J | Jackson Laboratory | IMSR_JAX:005584 | |
| Strain, strain background (mice) | B6.129(Cg)-*Gt(ROSA)26Sor*$^{tm4(ACTB-tdTomato,-EGFP)Luo}$/J | Jackson Laboratory | IMSR_JAX:007676 | |
| Commercial assay or Kit | Chromium Single Cell Next GEM 3′ Library & Gel Bead Kit v.3.1 | 10X Genomics | | |
| Commercial assay or Kit | RNAscope Multiplex Fluorescent Assay V2 | Bio-Techne | | |
| Antibody | Rat monoclonal to mouse SCA1 | 740450, BD Biosciences | RRID:AB_2740177 | Dilution: 1/200 |
| Antibody | Rabbit monoclonal to mouse SOX9 | ab185230, Abcam | RRID:AB_2715497 | Dilution: 1/1000 |
| Antibody | Rabbit polyclonal to mouse Osterix/Sp7 | ab22552, Abcam | RRID:AB_2194492 | Dilution: 1/200 |
| Antibody | Goat polyclonal to mouse Periostin | AF2955, R&D Systems | RRID:AB_664123 | Dilution: 1/400 |
| Antibody | Goat polyclonal to mouse PECAM1 | AF3628, Bio-Techne | RRID:AB_2161028 | Dilution: 1/200 |
| Antibody | Rat monoclonal to mouse CD45 | 552848, BD Biosciences | RRID:AB_394489 | Dilution: 1/200 |
| Software, algorithm | Seurat | https://github.com/satijalab/seurat; *Butler et al., 2024* | | |
| Software, algorithm | monocle3 | https://github.com/cole-trapnell-lab/monocle3; *Pliner et al., 2024* | | |
| Software, algorithm | CytoTrace | https://cytotrace.stanford.edu/ | | |
| Software, algorithm | EnrichR | https://maayanlab.cloud/Enrichr/ | | |
| Software, algorithm | SCENIC | https://github.com/aertslab/SCENIC; *Aibar, 2024* | | |
| Software, algorithm | STRING v11.5 database | https://string-db.org/ | | |
| Software, algorithm | CellChat | https://github.com/sqjin/CellChat; *Jin, 2024* | | |

### Mice

C57BL/6ScNj were obtained from Janvier Labs (France). *Prrx1*$^{Cre}$ (IMSR_JAX:005584) (*Logan et al., 2002*) and *Rosa26-mtdTomato-mEGFP* (*R26*$^{mTmG}$) (IMSR_JAX:007676) (*Muzumdar et al., 2007*) were obtained from Jackson Laboratory (Bar Harbor, ME). All SSPCs, including pSSPCs, are marked by GFP in *Prrx1*$^{Cre;}$ *R26*$^{mTmG}$ mice. Mice were bred in animal facilities at IMRB, Creteil, and kept in separated ventilated cages, in pathogen-controlled environment and ad libitum access to water and food. All procedures performed were approved by the Paris Est Creteil University Ethical Committee (#27176-2020091117563300, #19295-2019052015468705, #27181-202009141201846, #33818-2021110818301267, #33820-2021110819067229). Twelve-week-old males and females were mixed in experimental groups.

### Tibial fracture

Open non-stabilized tibial fractures were induced as previously described (*Perrin et al., 2021*). Mice were anesthetized with an intraperitoneal injection of ketamine (50 mg/mL) and Medetomidine (1 mg/

kg) and received a subcutaneous injection of buprenorphine (0.1 mg/kg) for analgesia. Mice were kept on a 37°C heating pad during anesthesia. The right hindlimb was shaved and sanitized. The skin was incised to expose the tibia and osteotomy was performed in the mid-diaphysis by cutting the bone. The wound was sutured, and the mice were revived with an intraperitoneal injection of atipamezole (1 mg/kg) and received two additional analgesic injections in the 24 hr following surgery. Mice were sacrificed at 3, 5, 7, or 14 days post-fracture.

## Nuclei extraction

Nuclei extraction protocol was adapted from *Santos et al., 2021* and *Martelotto, 2019*, and a detailed step-by-step protocol is available (*Perrin et al., 2024*). We generated four datasets for this study: uninjured periosteum, periosteum, and hematoma at days 3, 5, and 7 post-tibial fracture. The uninjured and day 3 post-fracture datasets were generated in duplicates. For uninjured periosteum, tibias from four mice were dissected free of muscle and surrounding tissues. The epiphyses were cut and the bone marrow flushed. The periosteum was scraped from the cortex using dissecting Chisel (10095-12, Fine Science Tools). For days 3, 5, and 7 post-fracture, injured tibias from 4 to 9 mice were collected and the surrounding tissues were removed. The activated periosteum was scraped and collected with the hematoma. Collected tissues were minced and placed 5 min in ice-cold Nuclei Buffer (NUC101, Merck) before mechanical nuclei extraction using a glass douncer. Extraction was performed by 20 strokes of pestle A followed by 5–10 of pestle B. Nuclei suspension was filtered, centrifuged, and resuspended in RNAse-free PBS (AM9624, Thermo Fisher Scientific) with 2% bovine serum albumin (A2153, Merck) and 0.2 U/µL RNAse inhibitor (3335399001, Roche). A second step of centrifugation was performed to reduce contamination by cytoplasmic RNA. Sytox AADvanced (S10349, Thermo Fisher Scientific) was added (1/200) to label nuclei and Sytox-AAD+ nuclei were sorted using Sony SH800.

## Single-nucleus RNA sequencing

The snRNA-seq libraries were generated using Chromium Single Cell Next GEM 3′ Library & Gel Bead Kit v.3.1 (10X Genomics) according to the manufacturer's protocol. Briefly, 10,000–20,000,000 nuclei were loaded in the 10X Chromium Controller to generate single-nucleus gel-beads in emulsion. After reverse transcription, gel-beads in emulsion were disrupted. Barcoded complementary DNA was isolated and amplified by PCR. Following fragmentation, end repair, and A-tailing, sample indexes were added during index PCR. The purified libraries were sequenced on a Novaseq (Illumina) with 28 cycles of read 1, 8 cycles of i7 index, and 91 cycles of read 2. Sequencing data were processed using the Cell Ranger Count pipeline, and reads were mapped on the mm10 reference mouse genome with intronic and exonic sequences.

## Filtering and clustering using Seurat

Single-nucleus RNAseq analyses were performed using Seurat v4.1.0 (*Stuart et al., 2019*; *Butler et al., 2018*) and RStudio v1.4.1717. Aligned snRNAseq datasets were filtered to retain only nuclei expressing between 200 and 5000 genes and expressing less than 2% of mitochondrial genes and 1.5% of ribosomal genes. Contamination from myogenic cells was removed from the analyses. After filtering, we obtained 1378 nuclei from uninjured periosteum, 1634 from day 3 post-fracture, 2089 from day 5 post-fracture, and 1112 from day 7 post-fracture. The replicates of the uninjured dataset were integrated using Seurat. The integrated dataset was regressed on cell cycle, mitochondrial, and ribosomal content, and clustering was performed using the first 15 principal components and a resolution of 0.5. SSPC/fibroblast and osteogenic cells were isolated and reclustered using the first 10 principal components and a resolution of 0.2. Uninjured, days 3, 5, and 7 datasets were integrated using Seurat. The integrated dataset was regressed on cell cycle, mitochondrial, and ribosomal content. Clustering was performed using the first 20 principal components and a resolution of 1.3. SSPC, IIFC, chondrogenic, and osteogenic clusters from the integration were isolated to perform subset analysis. The subset was reclustered using the first 15 principal components and a resolution of 0.6.

## Pseudotime analysis using monocle3

Monocle3 v1.0.0 was used for pseudotime analysis (*Cao et al., 2019*). Single-cell trajectories were determined using monocle3 default parameters. The starting point of the pseudotime trajectory was

determined as the cells from the uninjured dataset with the highest expression of stem/progenitor marker genes (*Ly6a, Cd34, Dpp4, Pi16*). Pseudotime values were added in the Seurat object as metadata and used with Seurat package.

### Differentiation state analysis using CytoTrace

To assess the level of differentiation of the cell clusters, we performed analyses using CytoTrace with the default parameters. CytoTrace scoring was plotted in violin plot and on the Seurat UMAP clustering.

### GO and Reactome and analyses

Reactome and GO analyses were performed using EnrichR (*Kuleshov et al., 2016*). All significant GO terms from upregulated genes in clusters 2–6 of the subset of SSPCs, IIFCs, osteoblasts, and chondrocytes were manually categorized. The five more significant terms of the Reactome analysis from the fibrogenic TFs of *Figure 6* are presented in *Supplementary file 3*.

### Single-cell regulatory network inference using SCENIC

Single-cell regulatory network inference and clustering (SCENIC) (*Aibar et al., 2017*) was used to infer transcription factor (TF) networks active in SSPCs, IIFCs, osteoblasts, and chondrocytes. Analysis was performed using recommended parameters using the packages SCENIC v1.3.1, AUCell v1.16.0, and RcisTarget v1.14 and the motif databases RcisTarget and GRNboost. SCENIC package was used to perform regulon-based tSNE clustering and identified population-specific regulons. Regulon activity values were added in the Seurat object as metadata and used with Seurat package for individual feature plots or lineage scores.

### Cell–cell interaction using CellChat

Cell communication analysis was performed using the R package CellChat (*Jin et al., 2021*), with default parameters on the complete fracture combined dataset and on the subset of SSPCs, IIFCs, osteoblasts, and chondrocytes.

### STRING network analyses

To assess protein–protein interaction network, we used the STRING v11.5 database (*Szklarczyk et al., 2021*). To assess interaction in the chondrogenic cluster, we performed the analysis on the chondro-core 1 and chondro-core 2 TFs identified in our analysis. For osteo-core analyses, we performed the analysis with osteo-core genes and the most significant interactions.

### Histology and immunofluorescence

Mouse samples were processed as previously described (*Perrin et al., 2021*). Tibias were collected and fixed in 4% PFA (sc-281692, CliniSciences) for 4 to 24h hr at 4°C. Then, samples were decalcified in 19% EDTA for 10 days (EU00084, Euromedex), cryoprotected in 30% sucrose (200-301-B, Euromedex) for 24 hr, and embedded in OCT. Samples were sectioned in 10-µm-thick sections. Cryosections were defrosted and rehydrated in PBS. For Safranin O staining, sections were stained with Weigert's solution for 5 min, rinsed in running tap water for 3 min, and stained with 0.02% Fast Green for 30 s (F7252, Merck), followed by 1% acetic acid for 30 s and Safranin O solution for 45 min (S2255, Merck). For immunofluorescence, sections were incubated 1 hr at room temperature in 5% serum, 0.25% Triton PBS before incubation overnight at 4°C with the following antibodies: rat monoclonal to mouse SCA1 (740450, RRID:AB_2740177, BD Biosciences), rabbit monoclonal to mouse SOX9 (ab185230, RRID:AB_2715497, Abcam), rabbit polyclonal to mouse Osterix/Sp7 (ab22552, RRID:AB_2194492, Abcam), goat polyclonal to mouse Periostin (AF2955, RRID:AB_664123, R&D Systems), goat polyclonal to mouse PECAM1 (AF3628, RRID:AB_2161028, Bio-Techne), and rat monoclonal to mouse CD45 (552848, RRID:AB_394489, BD Biosciences). Secondary antibody incubation was performed at room temperature for 1 hr. Slides were mounted with Fluoromount-G mounting medium with DAPI (00-4959-52, Life Technologies) and imaged using confocal microscopy (Carl Zeiss Microscopy GmbH).

### RNAscope in situ hybridization

The expression of *Pi16* and *Postn* was visualized using the RNAscope Multiplex Fluorescent Assay V2 (Bio-Techne). Tissues were processed as described above. 10-µm-thick sections were cut and

processed according to the manufacturer's protocol: 15 min of post-fixation in 4% PFA, ethanol dehydration, 10 min of $H_2O_2$ treatment, and incubation in ACD custom reagent for 30 min at 40°C. After hybridization and revelation, the sections were mounted under a glass coverslip with Prolong Gold Antifade (P10144, Thermo Fisher).

## Tissue dissociation and cell sorting

### Periosteal SSPC isolation

To isolate periosteal cells, uninjured tibias from *Prrx1^Cre^; R26^mTmG^* mice were collected, and all surrounding soft tissues were carefully removed. Epiphyses were embedded in low melting agarose and tibias were placed for 30 minutes at 37°C in digestion medium composed of PBS with 3mg/mL of collagenase B (C6885, Merck), 4mg/mL of Dispase (D4693, Merck), and 100U/mL of DNAse I (WOLS02007, Serlabo, France). After digestion, tibias were removed and the suspension was filtered, centrifuged, and resuspended.

### Injury-induced fibrogenic cell isolation

The fracture hematoma and the activated periosteum were collected from *Prrx1^Cre^; R26^mTmG^* mice 3 days post-fracture. Tissues were minced and digested at 37°C for 2 hr in DMEM (21063029, Life Technologies) with 1% Trypsin (15090046, Life Technologies) and 1% collagenase D (11088866001, Roche). Cells in suspension were removed every 20 min and the digestion medium was replaced. After 2 hr, the cell suspension was filtered, centrifuged, and resuspended.

### Cell sorting and transplantation

Digested cells were incubated 30 min with anti-SCA1-BV650 (BD Biosciences, 740450) or anti-CD146-BV605 (BD Biosciences, 740636). Cells were sorted using Influx Cell Sorter. Prior sorting, Sytox Blue (S34857, Thermo Fisher Scientific) was added to stain dead cells. We sorted living single GFP^+ SCA1^+ (pSSPCs excluding GFP^- endothelial cells and SCA1^- pericytes) and living single GFP^+ SCA1^- cells from the uninjured periosteum of *Prrx1^Cre^;R26^mTmG^* mice. For IIFCs, living single GFP^+ CD146^- cells were sorted, to eliminate CD146^+ pericytes. Cell transplantation was performed as described in **Perrin et al., 2021**. 30,000–45,000 sorted GFP^+ cells were embedded in Tisseel Prima fibrin gel, composed of fibrinogen and thrombin (3400894252443, Baxter S.A.S, USA), according to the manufacturer's instructions. Briefly, the cells were resuspended in 15 µL of fibrin (diluted at 1/4), before adding 15 µL of thrombin (diluted at 1/4) and mixing. The pellet was then placed on ice for at least 15 min for polymerization. The cell pellet was transplanted at the fracture site of wild-type mice at the time of fracture.

## In vitro colony-forming unit (CFU) assay

Prrx1-GFP^+ SCA1^+ and Prrx1-GFP^+ SCA1^- periosteal cells were isolated from *Prrx1^Cre^;R26^mTmG^* mice (male and female) as described above. Sorted cells were plated at a cell density of 2000 cells per 6-well plates in growth media consisting of MEMα supplemented with 20% FBS, 1% penicillin-streptomycin (Life Technology, Carlsbad, CA), and 0.01% bFGF (R&D, Minneapolis, MN). The medium was changed every 2–3 days. After 11 days, plates were washed twice with PBS, fixed with methanol, stained with 0.5% crystal violet diluted in 20% methanol, and colonies were manually counted.

## Statistical analyses

Statistical difference between CFU potential of SCA1+ and SCA1- cells was determined using Mann–Whitney test. Statistical difference between the number of activated regulons per cell was determined using one way-ANOVA followed by a post hoc test. p-Values <0.01 and <0.001 are reported as ** and ***, respectively.

## Acknowledgements

This work was supported by ANR-18-CE14-0033, ANR-21-CE18-007-01, NIAMS R01 AR072707, and R01 AR081671 to CC. SP, ME, and CG were supported by a PhD fellowship from Paris Cité University, Univ Paris-Est Creteil, and Fondation pour la Recherche Médicale, respectively. We thank O Ruckebusch and A Guigan from the Flow Cytometry platform at IMRB Institute, O Pellé from the

Flow Cytometry platform at Imagine Institute, and all the staff from the Imagine genomic core facility at Imagine Institute. We thank A Julien, S Protic, and Y Hachemi for technical assistance or advice.

## Additional information

### Funding

| Funder | Grant reference number | Author |
|---|---|---|
| Agence Nationale de la Recherche | ANR-18-CE14-0033 | Céline Colnot |
| Agence Nationale de la Recherche | ANR-21-CE18-007-01 | Céline Colnot |
| National Institute of Arthritis and Musculoskeletal and Skin Diseases | R01 AR072707 | Céline Colnot |
| National Institute of Arthritis and Musculoskeletal and Skin Diseases | R01 AR081671 | Céline Colnot |

The funders had no role in study design, data collection and interpretation, or the decision to submit the work for publication.

### Author contributions

Simon Perrin, Conceptualization, Formal analysis, Investigation, Visualization, Methodology, Writing – original draft; Maria Ethel, Cassandre Goachet, Investigation, Writing – review and editing; Vincent Bretegnier, Marine Luka, Cécile Masson, Investigation; Cécile-Aurore Wotawa, Formal analysis; Fanny Coulpier, Resources, Investigation; Mickael Ménager, Resources; Céline Colnot, Conceptualization, Supervision, Funding acquisition, Methodology, Writing – original draft, Project administration

### Author ORCIDs

Simon Perrin  https://orcid.org/0000-0003-3598-2617
Maria Ethel  https://orcid.org/0009-0002-0996-9427
Céline Colnot  https://orcid.org/0000-0001-8423-8718

### Ethics

All procedures performed were approved by the Paris Est Creteil University Ethical Committee, protocol apafis #27176-2020091117563300, #19295-2019052015468705, #27181-202009141201846, #33818-2021110818301267, #33820-2021110819067229.

Reviewer #1 (Public review): https://doi.org/10.7554/eLife.92519.3.sa1
Reviewer #2 (Public review): https://doi.org/10.7554/eLife.92519.3.sa2
Author response https://doi.org/10.7554/eLife.92519.3.sa3

## Additional files

### Supplementary files

- Supplementary file 1. Lists of genes used for lineage score analyses of murine snRNAseq.
- Supplementary file 2. Lists of the regulons composing the cores.
- Supplementary file 3. Top 5 terms from Reactome analysis on fibro-core regulons.
- MDAR checklist

### Data availability

Single nuclei RNAseq datasets are deposited in the Gene Expression Omnibus (GSE234451). The integrated dataset was deposited on USCS Cell Browser for easy use: (https://fracture-repair-atlas.cells.

ucsc.edu). This paper does not report original code. Further information and requests for resources and reagents should be directed to and will be fulfilled by the lead contact, Céline Colnot (celine.colnot@inserm.fr).

The following datasets were generated:

| Author(s) | Year | Dataset title | Dataset URL | Database and Identifier |
|---|---|---|---|---|
| Perrin S, Colnot C | 2024 | Single nuclei RNAseq of the periosteal response to bone fracture in mice | http://www.ncbi.nlm.nih.gov/geo/query/acc.cgi?acc=GSE234451 | NCBI Gene Expression Omnibus, GSE234451 |
| Perrin S | 2024 | Fracture Repair Atlas | https://fracture-repair-atlas.cells.ucsc.edu | USCS Cell Browser, fracture-repair-atlas |

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
