## [Editor Report · eLife Assessment]

This **fundamental** study generated a single-cell atlas of mouse periosteal cells under both steady-state and fracture healing conditions to address the knowledge gap regarding cellular composition of the periosteum and their responses to injury. Based on **convincing** transcriptome analyses and experimental validation, the authors identified the injury induced fibrogenic cell (IIFC) as a characteristic cell type appearing in the bone regeneration process and proposed that the IIFC is a progenitor undergoing osteochondrogenic differentiation. This study will provide a significant publicly accessible dataset to reexamine the expression of the reported periosteal stem and progenitor cell markers.

---

## [Referee Report · Reviewer #1 (Public review)]

This study delineates an important set of uninjured and injured periosteal snRNAseq data that provides an overview of periosteal cell responses to fracture healing. The authors also took additional steps to validate some of the findings using immunohistochemistry and transplantation assays. This study will provide a valuable publicly accessible dataset to reexamine the expression of the reported periosteal stem and progenitor cell markers.

Strengths:

(1) This is the first single-nuclei atlas of periosteal cells that are obtained without enzymatic cell dissociation or targeted cell purification by FACS. This integrated snRNAseq dataset will provide additional opportunities for the community to revisit the expression of many periosteal cell markers that have been reported to date.

(2) The authors delved further into the dataset using cutting-edge algorithms, including CytoTrace, SCENIC, Monocle, STRING and CellChat, to define potential roles of identified cell populations in the context of fracture healing. These additional computation analyses generate many new hypotheses regarding periosteal cell reactions.

(3) The authors also sought to validate some of the computational findings using immunohistochemistry and transplantation assays to support the conclusion.

Weaknesses:

(1) The current snRNAseq datasets contain only a small number of nuclei (1,189 nuclei at day 0, 6,213 nuclei day 0-7 combined). It is possible that these datasets are underpowered to discern subtle biological changes in skeletal stem/progenitor cell populations during fracture healing.

(2) POSTN is expressed in the cambium layer of the periosteum without fracture. The current data do not exclude the possibility that these pre-existing POSTN+ cells are the main responder of fracture healing.

---

## [Referee Report · Reviewer #2 (Public review)]

Summary:

The authors described cell type mapping was conducted for both WT and fracture types. Through this, unique cell populations specific to fracture conditions were identified. To determine these, the most undifferentiated cells were initially targeted using stemness-related markers and CytoTrace scoring. This led to the identification of SSPC differentiating into fibroblasts. It was observed that the fibroblast cell type significantly increased under fracture conditions, followed by subsequent increases in chondrocytes and osteoblasts.

Strengths:

This study presented the injury-induced fibrogenic cell (IIFC) as a characteristic cell type appearing in the bone regeneration process and proposed that the IIFC is a progenitor undergoing osteochondrogenic differentiation.

Comments on revised version:

The authors have thoroughly addressed the reviewer's comments and have conducted additional experiments.

---

## [Author Response]

The following is the authors’ response to the original reviews.

**Public Reviews:**

**Reviewer #1 (Public Review):**
This study delineates an important set of uninjured and injured periosteal snRNAseq data that provides an overview of periosteal cell responses to fracture healing. The authors also took additional steps to validate some of the findings using immunohistochemistry and transplantation assays. This study will provide a valuable publicly accessible dataset to reexamine the expression of the reported periosteal stem and progenitor cell markers.Strengths:(1) This is the first single-nuclei atlas of periosteal cells that are obtained without enzymatic cell dissociation or targeted cell purification by FACS. This integrated snRNAseq dataset will provide additional opportunities for the community to revisit the expression of many periosteal cell markers that have been reported to date.(2) The authors delved further into the dataset using cutting-edge algorithms, including CytoTrace, SCENIC, Monocle, STRING, and CellChat, to define the potential roles of identified cell populations in the context of fracture healing. These additional computation analyses generate many new hypotheses regarding periosteal cell reactions.(3) The authors also sought to validate some of the computational findings using immunohistochemistry and transplantation assays to support the conclusion.Weaknesses:(1) The current snRNAseq datasets contain only a small number of nuclei (1,189 nuclei at day 0, 6,213 nuclei on day 0-7 combined). It is unclear if the number is sufficient to discern subtle biological processes such as stem cell differentiation.

We analyzed a total of 6,213 nuclei from uninjured periosteum and fracture calluses at 3 stages of bone healing. We were able to describe 11 distinct cell populations, revealing the diversity of cell populations in uninjured periosteum and post-injury, including rare cell types in the fracture environment such Schwann cells, adipocytes and pericytes. The number of nuclei was sufficient to perform extensive analysis using a combination of cutting-edge algorithms. We agree that more nuclei would allow more in-depth analyses of cell fate transitions and rare populations, such as pericytes and Schwann cells. However, we concentrated here on SSPC/fibrogenic cells that are well represented in our dataset. Our study robustness is also reinforced by the analysis of 4 successive time points to define the SSPC/fibrogenic cell trajectories. Our validations using immunohistochemistry and transplantation assays also confirmed that our dataset is sufficient to define cell trajectories. There is no clear consensus on the number of cells needed to perform sc/snRNAseq analyses, as it depends on the cell types analyzed and the fold changes in gene expression. Previously reported single cell datasets containing a lower number of cells reached major conclusions including SSPC identification, cell differentiation trajectories and differential gene expression (658 cells in (Debnath et al. 2018), 300 in (Ambrosi et al. 2021), around 175 in (Remark et al. 2023).)

(2) The authors' designation of Sca1+CD34+ cells as SSPCs is not sufficiently supported by experimental evidence. It will be essential to demonstrate stem/progenitor properties of Sca1+CD34+ cells using independent biological approaches such as CFU-F assays. In addition, the putative lineage trajectory of SSPCs toward IIFCs, osteoblasts, and chondrocytes remains highly speculative without concrete supporting data.

We performed additional analyses to further support that SCA1+ SSPCs display stem/progenitor properties. We performed CFU assays with Prx1-GFP+ SCA1+ and Prx1-GFP+ SCA1- periosteal cells (Figure 2F-G). We showed that Prx1-GFP+ SCA1+ display significant increased CFU potential compared to Prx1-GFP+ SCA1- cells. In addition, we isolated and transplanted Prx1-GFP+ SCA1+ and Prx1-GFP+ SCA1- periosteal cells at the fracture site of wild-type mice (Figure 2H). Only SCA1+ cells contributed to the callus formation, reinforcing that SCA1+ cells are the SSPC population mediating bone repair.

The differentiation trajectory of SSPCs presented in our study is supported by a combination of bioinformatic analyses and in vivo validation:

- snRNAseq allowed us to identify the different populations in the uninjured periosteum. In silico, in vitro and in vivo analyses all point to SCA1+ cells as the SSPC population (Fig 2EG).

- At day 3 post-fracture, we did not detect SCA1+ cells in the callus (Fig 4 – Supplementary figure 2). Instead, we observed the appearance of a new population, IIFCs. This population clustered along SSPCs and pseudotime analyses indicate that SSPCs can differentiate into IIFCs (Fig 5B). We confirmed the ability of SCA1+ pSSPCs to form IIFCs, by grafting them in the fracture callus and assessing their fibrogenic fate at day 5 post-fracture (Fig 6B).

- In silico, we observed that IIFCs clustered along osteogenic and chondrogenic cells. The pseudotime trajectory suggests that IIFCs can differentiate into both lineages (Fig 5B-C). This is coherent with the progressive expression of osteochondrogenic genes observed in IIFCs (Fig 5C, Fig 8A, C, E). In vivo, we observed the progressive expression of Runx2 and Sox9 by IIFCs undergoing differentiation (Fig 6A). We now show that IIFCs are not undergoing apoptosis, indicating that these cells further differentiate (Fig 7 – Supplementary figure 2). To functionally assess the osteochondrogenic potential of IIFCs, we used transplantation assay and showed that Prx1-GFP+ IIFCs isolated from day 3 post-fracture form cartilage and bone when transplanted at the fracture site of wild-type mice (Fig 6C).

We would like to insist on the robustness of the bioinformatic analyses performed in our study. First, we used datasets from different time points post-fracture to capture the true temporal progression of cell populations in the fracture callus. We used a large combination of tools shown to be reliable in many studies (Julien et al. 2021; Matsushita et al. 2020; Debnath et al. 2018; Baccin et al. 2020; Junyue Cao et al. 2019; Zhong et al. 2020), and all tools converge in the same trajectory. To further show the relevance of pseudotime in our model, we illustrated the distribution of the cell populations by time point (Fig. 5D). We can observe a parallel between the time points and the pseudotime, reinforcing that the pseudotime trajectory reflects the timing of SSPC differentiation. Overall, the combined in silico, in vitro and in vivo analyses support that SCA1+ Pi16+ cells are the periosteal SSPC population, specifically represented in the uninjured dataset. In response to bone fracture, these SSPCs give rise to IIFCs that are specifically represented in the intermediate stages (days 3 and 5) prior to osteochondrogenic differentiation.

(3) The designation of POSTN+ clusters as injury-induced fibrogenic cells (IIFCs) is not fully supported by the presented data. The authors' snRNAseq datasets (Figure 1d) demonstrate that there are many POSTN+ cells prior to injury, indicating that POSTN+ cells are not specifically induced in response to injury. It has been widely recognized that POSTN is expressed in the periosteum without fracture. This raises a possibility that the main responder of fracture healing is POSTN+ cells, not SSPCs as they postulate. The authors cannot exclude the possibility that Sca1+CD34+ cells are mere bystanders and do not participate in fracture healing.

IIFCs are a population of cells that express high levels of ECM related genes, including *Postn*, *Aspn* and collagens. We did not claim that *Postn* expression is specific to IIFCs. While *Postn* is detected in the uninjured periosteum, snRNAseq analyses and RNAscope experiments showed that the expression of *Postn* is limited to a small number of cells in the cambium layer of the periosteum (Fig 4B , Figure 4 – Supplementary figure 1B). These *Postn*-expressing cells in the uninjured periosteum are not SSPCs, as they do not co-express/co-localize with Pi16+ and SCA1+ cells detected in the fibrous layer (Fig4, Figure 4– Supplementary figure 1A, Figure 6-Supplementary figure 1). These *Postn*-expressing cells are undergoing osteogenic differentiation as shown by the correlation between *Runx2* and *Postn* expression (Fig. 4 – Supplementary Figure 1C). After fracture, we observed a strong increase in ECM-related gene expression and specifically in the IIFC population. We now show the strong increase of *Postn* expression after injury (Fig. 4 – Supplementary Figure 1D-E, Figure 6-Supplementary figure 1E).

As mentioned in our response above, we now show that SCA1+ cells form cartilage and bone after fracture, while SCA1- cells (including the POSTN+ population) from the uninjured periosteum did not contribute. These data reveal that SCA1+ CD34+ cells are the main SSPC population mediating bone healing and that POSTN+ IIFCs are a transient stage of SSPC differentiation. We added the following text to the result section: “*Pi16*-expressing SSPCs are located within the fibrous layer, while we observed few POSTN+ cells in the cambium layer (Fig. 4 – Supplementary Fig. 1A). *Postn* expression is weak in uninjured periosteum and is limited to differentiating cells. *Postn* expression is strongly increased in response to fracture, specifically in IIFCs (Fig. 4 – Supplementary Fig. 1B-E). “

(4) Detailed spatial organization of Sca1+CD34+ cells and POSTN+ cells in the uninjured periosteum with respect to the cambium layer and the fibrous layer is not demonstrated.

We performed RNAscope experiments to locate *Pi16*-expressing and *Postn*-expressing cells in the uninjured periosteum. We observed that *Pi16*-expressing cells are in the external fibrous layer of the periosteum while *Postn*-expressing cells are located along the cortex in the cambium layer. The data are added in Fig 4B and Fig. 4- Supplementary Figure 1 and mentioned in the result section “*Pi16*-expressing SSPCs were located within the fibrous layer, while *Postn*-expressing cells were found in the cambium layer and corresponded to Runx2-expressing osteogenic cells (Fig. 4 – Supplementary Fig. 1A-C).”.

(5) Interpretation of transplantation experiments in Figure 5 is not straightforward, as the authors did not demonstrate the purity of Prx1Cre-GFP+SCA1+ cells and Prx1Cre-GFP+CD146- cells to pSSPCs and IIFCs, respectively. It is possible that these populations contain much broader cell types beyond SSPCs or IIFCs.

We agree with the reviewer that our methodology for cell transplantation required more justification and validation. We decided to use a transgenic mouse line to be able to trace the cells in vivo after grafting. *Prx1* marks limb mesenchyme during development and the *Prx1Cre* mouse model allows to label all SSPCs contributing to callus formation. Therefore, we used *Prx1Cre*, *R26mTmG* mice as donors for SSPCs and IIFCs isolation (Duchamp de Lageneste et al. 2018; Logan et al. 2002). *Prx1* does not mark immune and endothelial cells but can label pericytes and fibroblastic populations (Duchamp de Lageneste et al. 2018; Logan et al. 2002; Julien et al. 2021). In the uninjured periosteum, SCA1 (*Ly6a*) is only expressed by SSPCs and endothelial cells (Fig 3-Supplementary figure 2, Fig 6-Supplementary figure 1). We sorted GFP+ SCA1+ cells from uninjured periosteum of *Prx1Cre*, *R26mTmG* mice to isolate only SSPCs and excluding endothelial cells and pericytes. For IIFCs, we isolated cells at day 3 post-fracture, as in our snRNAseq data, we detected IIFCs but no SSPCs, chondrocytes or osteoblasts at this stage of repair. To eliminate Prx1-derived pericytes, we sorted GFP+CD146- cells, as CD146 is specifically expressed by pericytes. We added Figure 6-supplementary Figure 1 to better illustrate the expression of *Prx1*, SCA1 (*Ly6a*) and CD146 (*Mcam*) in the uninjured and day 3 post-fracture datasets. We further demonstrate the purity of SSPCs and IIFCs isolation by qPCR on sorted GFP+ SCA1+ cells from uninjured periosteum and GFP+ CD146- cells from day 3 post-fracture periosteum and hematoma and confirmed the absence of contamination by other cell populations (Figure 6-Supplementary figure 1E). We made the following changes in the text: “To functionally validate the steps of pSSPC activation, we isolated SCA1+ GFP+ pSSPCs from *Prx1Cre; R26mTmG* mice, excluding endothelial cells, and grafted them at the fracture site of wild-type hosts” and “we isolated GFP+ CD146- from the fracture callus of *Prx1Cre; R26mTmG* mice at day 3 post fracture, that correspond to IIFCs without contamination by pericytes (CD146+ cells) (Fig. 6C, Figure 6 – Supplementary Fig.1).

**Reviewer #2 (Public Review):**
Summary:The authors described cell type mapping was conducted for both WT and fracture types. Through this, unique cell populations specific to fracture conditions were identified. To determine these, the most undifferentiated cells were initially targeted using stemness-related markers and CytoTrace scoring. This led to the identification of SSPC differentiating into fibroblasts. It was observed that the fibroblast cell type significantly increased under fracture conditions, followed by subsequent increases in chondrocytes and osteoblasts.Strengths:This study presented the injury-induced fibrogenic cell (IIFC) as a characteristic cell type appearing in the bone regeneration process and proposed that the IIFC is a progenitor undergoing osteochondrogenic differentiation.Weaknesses:This study endeavored to elucidate the role of IIFC through snRNAseq analysis and in vivo observation. However, such validation alone is insufficient to confirm that IIFC is an osteochondrogenic progenitor, and additional data presentation is required.

As mentioned in the response to Reviewer 1, the differentiation trajectory of SSPCs presented in our study is supported by a combination of bioinformatic analyses and in vivo validation:

- snRNAseq allowed us to identify the different populations in the uninjured periosteum. In silico, in vitro and in vivo analyses altogether showed that SCA1+ cells are the SSPC population (Fig 2E-G).

- At day 3 post-fracture, we did not detect SCA1+ cells in the callus (Fig 4 – Supplementary figure 2). Instead, we observed the appearance of a new population, IIFCs. This population clustered along SSPCs and pseudotime analyses indicate that SSPCs can differentiate into IIFCs (Fig 5B). We confirmed the ability of SCA1+ SSPCs to form IIFCs, by grafting them in the fracture callus and assessing their fate at day 5 post-fracture (Fig 6B).

- In silico, we observed that IIFCs clustered along osteogenic and chondrogenic cells. The pseudotime trajectory suggests that IIFCs can differentiate into both lineages (Fig 5B-C). This is coherent with the progressive expression of osteochondrogenic genes observed in IIFCs (Fig 5C, Fig 8A, C, E). In vivo, we observed the progressive expression of Runx2 and Sox9 by IIFCs undergoing differentiation (Fig 6A). We now show that IIFCs are not undergoing apoptosis, indicating that these cells further differentiate (Fig 7 – Supp 2). To functionally assess the osteochondrogenic potential of IIFCs, we used transplantation assay and showed that Prx1-GFP+ IIFCs from day 3 post-fracture form cartilage and bone when transplanted at the fracture site of wild-type mice (Fig 6C).

We would like to insist on the robustness of the bioinformatic analyses performed in our study. First, we used datasets from different time points post-fracture to capture the true temporal progression of cell populations in the fracture callus. We used a large combination of tools shown to be reliable in many studies (Julien et al. 2021; Matsushita et al. 2020; Debnath et al. 2018; Baccin et al. 2020; Junyue Cao et al. 2019; Zhong et al. 2020), and all tools converge in the same trajectory. To further show the relevance of pseudotime in our model, we illustrate the distribution of the cell populations by time point (Fig. 5D). We can observe a parallel between the time points and the pseudotime, reinforcing that the pseudotime trajectory reflects the timing of SSPC differentiation. Overall, the combined in silico, in vitro and in vivo analyses strongly support that SCA1+ Pi16+ cells are the periosteal SSPC population, specifically represented in the uninjured dataset. In response to bone fracture, these SSPCs give rise to IIFCs that are specifically represented in the intermediate stages (days 3 and 5) prior to osteochondrogenic differentiation.

We made the following changes in the text:

- Line 81-87: “We performed in vitro CFU assays with sorted GFP+SCA1+ and GFP+SCA1- cells isolated from the periosteum of Prx1Cre; R26mTmG mice, as Prx1 labels all SSPCs contributing to the callus formation1. Prx1-GFP+ SCA1+ showed increased CFU potential, confirming their stem/progenitor property (Fig 2F-G). Then, we grafted Prx1GFP+ SCA1+ et Prx1-GFP+ SCA1- periosteal cells at the fracture site of wild-type mice. Only SCA1+ cells formed cartilage and bone after fracture indicating that SCA1+ cells correspond to periosteal SSPCs with osteochondrogenic potential (Fig 2H).”

- Line 120-122: “We did not detect Pi16-expressing SPPCs, consistent with the absence of cells expressing SSPC markers in day 3 snRNAseq dataset compared to uninjured periosteum (Fig. 4 – Supplementary Figure 2).”

- Line 170-172: “Only a small subset of IIFCs undergo apoptosis, further supporting that IIFCs are maintained in the fracture environment giving rise to osteoblasts and chondrocytes (Fig. 7 – Supplementary Figure 2).”

- Line 277-278: “Following this unique fibrogenic step, IIFCs do not undergo cell death but undergo either osteogenesis or chondrogenesis”

- Line 281-283: “During bone repair, this initial fibrogenic process is an integral part of the SSPC differentiation process, and a transitional step prior to osteogenesis and chondrogenesis.”

**Reviewer #3 (Public Review):**
In this manuscript, the authors explored the transcriptional heterogeneity of the periosteum with single nuclei RNA sequencing. Without prior enrichment of specific populations, this dataset serves as an unbiased representation of the cellular components potentially relevant to bone regeneration. By describing single-cell cluster profiles, the authors characterized over 10 different populations in combined steady state and post-fracture periosteum, including stem cells (SSPC), fibroblast, osteoblast, chondrocyte, immune cells, and so on. Specifically, a developmental trajectory was computationally inferred using the continuum of gene expression to connect SSPC, injury-induced fibrogenic cells (IIFC), chondrocyte, and osteoblast, showcasing the bipotentials of periosteal SSPCs during injury repair. Additional computational pipelines were performed to describe the possible gene regulatory network and the expected pathways involved in bone regeneration. Overall, the authors provided valuable insights into the cell state transitions during bone repair and proposed sets of genes with possible involvements in injury response.While the highlights of the manuscript are the unbiased characterization of periosteal composition, and the trajectory of SSPC response in bone fracture response, many of the conclusions can be more strongly supported with additional clarifications or extensions of the analysis.(1) As described in the method section, both the steady-state data and full dataset underwent integration before dimensional reduction and clustering. It would be appreciated if the authors could compare the post-integration landscapes of uninjured cells between steady state and full dataset analysis. Specifically, fibroblasts were shown in Figure 1C and 1E, and such annotations did not exist in Figure 2B. Will it be possible that the original 'fibroblasts' were part of the IIFC population?

As suggested, we now identified the fibroblast population from the uninjured periosteum in the integration of datasets from all time points (Figure 5B and Fig. 5 – Supplementary Figure 2). We identified 4 fibroblast populations in the uninjured periosteum: Luzp2+, Cldn1+, Hsd11b1+ and Csmd1+ fibroblasts. Luzp2+ and Cldn1+ fibroblasts are clustering distinctly from the other populations in the integrated dataset. Hsd11b1+ fibroblasts blend with SSPCs and IIFCs in the integrated dataset probably due to the low cell number. Finally, Csmd1+ fibroblasts are clustering at the interface between SSPCs and IIFCs likely because they correspond to differentiating cells both in the uninjured periosteum and in response to fracture. We modified the resolution of clustering in our subset dataset, in order to represent Luzp2+ and Cldn1+ fibroblasts as an isolated cluster (Figure 5B, cluster 10). In addition, both pseudotime (Fig. 5B) and gene regulatory network analyses (Fig. 7D), show that the fibroblast populations are distinct from the activation trajectory of SSPCs. We added the following sentence to the text “Fibroblasts from uninjured periosteum (Hsd11b1+, Cldn1+ and Luzp2+ cells corresponding to cluster 10 of Fig. 5B) clustered separately from the other populations, suggesting the absence of their contribution to bone healing.”

(2) According to Figure 2, immune cells were taking a significant abundance within the dataset, specifically during days 3 & 5 post-fracture. It will be interesting to see the potential roles that immune cells play during bone repair. For example, what are the biological annotations of the immune clusters (B, T, NK, myeloid cells)? Are there any inflammatory genes or related signals unregulated in these immune cells? Do they interact with SSPC or IIFC during the transition?

In this manuscript, we report the overall dataset and focused our analyses on the response of SSPCs to injury and their differentiation trajectories. We did not include detailed analyses of the immune cell populations, that are out of scope of this manuscript and are part of another study (Hachemi et al, biorxiv, 2024)

(3) The conclusion of Notch and Wnt signaling in IIFC transition was not sufficiently supported by the analysis presented in the manuscript, which was based on computational inferences. It will be great to add in references supporting these claims or provide experimental validations examining selected members of these pathways.

The role of Wnt and Notch in bone repair has been widely studied and both signaling pathways are known to be regulators of SSPCs differentiation (Lee et al. 2021; Matthews et al. 2014; Novak et al. 2020; Wang et al. 2016; Kraus et al. 2022; Dishowitz et al. 2012; Junjie Cao et al. 2017; Matsushita et al. 2020; Steven Minear et al. 2010; Steve Minear et al. 2010; Kang et al. 2007; Komatsu et al. 2010). It was previously shown that Notch inactivation at early stages of repair leads to bone non-union while Notch inactivation in chondrocytes and osteoblasts does not significantly affect healing, confirming its role in SSPC differentiation before osteochondral commitment (Wang et al. 2016). Wnt was shown to be a critical driver of osteogenesis (Matsushita et al. 2020; Steve Minear et al. 2010; Steven Minear et al. 2010; Kang et al. 2007; Komatsu et al. 2010), as Wnt inhibition alters bone formation and Wnt overactivation increases bone formation (Pinzone et al. 2009; Balemans et Van Hul 2007). The role of Wnt is specific to osteogenic engagement as Wnt inhibition promotes chondrogenesis (Hsieh et al. 2023; C.-L. Wu et al. 2021; Ruscitto et al. 2023). A study by Lee et al. recently confirmed the successive activation and crosstalk of Notch and Wnt pathways during osteogenic differentiation of SSPCs during bone healing (Lee et al. 2021). They showed a peak of Notch activation at day 3 post-injury followed by a progressive decrease that parallels an increase of Wnt signaling inducing osteogenic differentiation. These studies correlate with the sequential activation of Notch and Wnt observed in our snRNAseq analyses. Our analyses now reveal how this sequential activation of Notch and Wnt relates to the fibrogenic and osteogenic phase of SSPC differentiation respectively. We clarified this in the discussion and added the references above to support our claims.

**Recommendations for the authors:**

**Reviewer #1 (Recommendations For The Authors):**
(1) The manuscript is well-written overall. However, the authors often oversimplify outcomes and overstate the results. Some of the statements (delineated below) need to be recalibrated to be in line with the presented data.

In addition to the suggested conclusions, we also toned down the following ones to avoid overstating our results :

Line 24: suggesting a crucial paracrine role of this transient IIFC population

Line 227: suggesting their central role in mediating cell interactions after fracture

line 243: IIFCs produce paracrine factors that can regulate SSPCs

- Line 77 (86): The authors should add "might" before "correspond to".

We provided new sets of data including CFU experiments and transplantation assay to reinforce our conclusion. We replaced “correspond to” by “encompass”

- Line 102: SSPCs are obviously not "absent" in day 3 snRNAseq (Figure 2d). The percentage dropped (only) 75%, according to Figure 2e, which is far from disappearance. Overall, immunohistochemical staining is often dichotomous with snRNAseq designations. The authors should more carefully describe the results.

We agree that this comment may not reflect the data shown as we observe a strong decrease in the percentage of cells in SSPC clusters, but still detect few cells in the SSPC clusters. However, when we looked at the presence of SCA1+ Pi16+ cells at different time points, we confirmed the absence of cells expressing SSPC signature genes (SCA1, Pi16, Cd34) at day 3 injury. Due to the clustering resolution of the combined integration, some cells in the SSPC clusters might not be SCA1+ Pi16+. We now show these results in Fig. 4 – Supplementary Figure 2. We changed the text accordingly (line 120): “We did not detect *Pi16*-expressing SPPCs, consistent with the absence of cells expressing SSPC markers in the day 3 snRNAseq dataset compared to uninjured periosteum (Fig. 4 – Supplementary Figure 2)”.

- Line 134: The authors need to clearly state that GFP+IIFCs were isolated based on Prx1CreGFP+CD146-. The authors did not clearly demonstrate the relationship between POSTN+ cells and CD146- cells, which poses concerns about the interpretation of transplantation experiments.

As mentioned above in response to reviewer 1-public review, we have clarified and provided additional information on our strategy to isolate SSPCs and IIFCs. We used the *Prx1Cre; R26mTmG* mice to mark all SSPCs and their derivatives with the GFP reporter in order to trace these populations after cell grafting. In the uninjured periosteum, SCA1 (*Ly6a*) is only expressed by SSPCs and endothelial cells. We sorted GFP+SCA1+ cells to exclude endothelial cells. For IIFCs, we isolated cells at day 3 post-fracture, as in our snRNAseq data, we detect IIFCs but no SSPCs, chondrocytes or osteoblasts at this time point. However, we also detected pericytes that can be Prx1-derived. To eliminate potential pericyte contamination, we sorted GFP+ CD146- cells, as CD146 is specifically expressed by pericytes. We added Figure 6-supplementary Figure 1 to better illustrate the expression of *Prx1*, SCA1 (*Ly6a*) and CD146 (*Mcam*) in the uninjured and day 3 post-fracture datasets. We further demonstrate the purity of SSPCs and IIFCs isolation by qPCR on sorted GFP+ SCA1+ cells from uninjured periosteum and GFP+ CD146- cells from day 3 postfracture periosteum and hematoma and confirmed the absence of contamination by other cell populations (Figure 6-Supplementary figure 1E). We made the following changes in the text (line 153): “To functionally validate the steps of pSSPC activation, we isolated SCA1+ GFP+ pSSPCs from *Prx1Cre; R26mTmG* mice, excluding endothelial cells, and grafted them at the fracture site of wild-type hosts” and “we isolated GFP+ CD146- from the fracture callus of *Prx1Cre; R26mTmG* mice at day 3 post fracture, that correspond to IIFCs without contamination by pericytes (CD146+ cells) (Fig. 6C, Figure 6 – Supplementary Fig.1).

- Line 211: It is obvious from Figure 8F that ligand expression was not "specific" to the IIFC phase.The data only shows a slight enrichment of ligand score.

We corrected the text by “ligand expression was increased during the IIFC phase”.

(2) Some of the computational predictions are incongruent with the known lineage trajectory. For example, in vivo lineage tracing experiments, including but not limited to, PLoS Genet. 2014. 10:e1004820, demonstrate that some of the chondrocytes within fracture callus can differentiate into osteoblasts. This is incompatible with the authors' conclusion that osteoblasts and chondrocytes represent two different terminal stages of cell differentiation in fracture healing. How do the authors reconcile this apparent inconsistency?

In this manuscript, we generated datasets corresponding to the initial stages of bone repair until day 7 post-injury. Therefore, our analyses encompass SSPC activation stages and engagement into osteogenesis and chondrogenesis. The results show that a portion of osteoblasts in the fracture callus are differentiating directly from IIFC via intramembranous ossification. The reviewer is correct to mention that osteoblasts have also been shown to derive from transdifferentiation of chondrocytes, which occurs at later stages of repair during the active phase of endochondral ossification (Julien et al. 2020; Aghajanian et Mohan 2018; Zhou et al. 2014; Hu et al. 2017). This process of chondrocyte to osteoblast transdifferentiation is not represented in our integrated dataset and may require adding later time points. However, when we analyzed the days 5 and 7 datasets independent of days 0 and 3, we were able to identify a cluster of hypertrophic chondrocytes (expressing *Col10a1*) connecting the clusters of chondrocytes and osteoblasts. This suggests that in this cluster, hypertrophic chondrocytes are undergoing transdifferentiation into osteoblasts as shown in the Author response image 1. Additional time points are needed in a future study to perform in depth analyses of chondrocyte transdifferentiation.

**Author response image 1. sa3fig1:** Periosteum-derived chondrocytes undergo cartilage to bone transformation. A. UMAP projection of the subset of SSPCs, IIFCs, osteoblasts and chondrocytes in the integration of days 5 and 7 post-fracture datasets. B. Feature plots of *Acan*, *Col10a1* and *Ibsp* expression. C. UMAP projection separated by time points. D. Percentage of cells in the hypertrophic/differentiating chondrocyte cluster.

(3) The authors did not cite some of the studies that described the roles of Notch signaling in fracture healing, for example, J Bone Miner Res. 2014. 29:1283-94. The authors should test the specificity of Notch signaling activities to IIFCs (POSTN+ cells) in vivo.

The role of Notch in the activation of SSPCs during bone repair has been investigated in several studies (Lee et al. 2021; Matthews et al. 2014; Novak et al. 2020; Wang et al. 2016; Kraus et al. 2022; Dishowitz et al. 2012; Junjie Cao et al. 2017). Notch dynamic was previously described with a peak at day 3 post-injury before a reduction when cells engage in osteogenesis and chondrogenesis (Lee et al. 2021; Dishowitz et al. 2012; Matthews et al. 2014). Notch plays a role in the early steps of SSPC activation prior to osteochondral differentiation as Notch inactivation in chondrocytes and osteoblasts does not affect bone repair (Wang et al. 2016). We added the references listed above to emphasize the correlation between our results and previous reports on the role of Notch and made changes in the discussion.

**Reviewer #2 (Recommendations For The Authors):**
Suggestions(1) This research utilized snRNA seq for the basic hypothesis formation; however, the number of nuclei acquired was quite limited. Therefore, please explain the rationale for employing snRNA seq instead of scRNA seq, which includes cytoplasm, and additionally provide the markers used for cell type mapping in the scRNA analysis.

As mentioned in our response to reviewer #1 above, we analyzed a total of 6,213 nuclei from uninjured periosteum and fracture calluses at 3 stages of bone healing. We were able to describe 11 distinct cell populations including rare cell types in the fracture environment such Schwann cells, adipocytes and pericytes. The number of nuclei was sufficient to perform extensive analysis using a combination of cutting-edge algorithms. We agree that more nuclei would allow more indepth analyses of cell fate transitions and rare populations, such as pericytes and Schwann cells. However, we concentrated here on SSPC/fibrogenic cell that are well represented in our dataset. Our study robustness is also reinforced by the analysis of 4 successive time points to define the SSPC/fibrogenic cell trajectories. Our validations using immunohistochemistry and transplantation assays also confirmed that our dataset is sufficient to define cell trajectories. There is no clear consensus on the number of cells needed to perform scRNAseq analyses, as it depends on the cell types analyzed and the fold changes in gene expression. Previously reported single cell datasets containing a lower number of cells reached major conclusions including SSPC identification, cell differentiation trajectories and differential gene expression (658 cells in(Debnath et al. 2018), 300 in (Ambrosi et al. 2021) around 175 in(Remark et al. 2023))

Several studies have shown that snRNAseq provide data quality equivalent to scRNAseq in terms of cell type identification, number of detected genes and downstream analyses (Selewa et al. 2020; Wen et al. 2022; Ding et al. 2020; H. Wu et al. 2019; Machado et al. 2021). While, snRNAseq do not allow the detection of cytoplasm RNA, there is several advantages in using this technique:

(1) better representation of the cell types. To perform scRNAseq, a step of enzymatic digestion is needed. This usually leads to an overrepresentation of some cell types loosely attached to the ECM (immune cells, endothelial cells) and a reduced representation of cell types strongly attached to the ECM, such as chondrocytes and osteoblasts. In addition, large or multinucleated cells like hypertrophic chondrocytes and osteoclasts are too big to be sorted and encapsidated using 10X technology. Here, we optimized a protocol to mechanically isolate nuclei from dissected tissues that allows us to capture the diversity of cell types in periosteum and fracture callus.

(2) higher recovery of nuclei. We performed both isolation of cells and nuclei from periosteum in our study and observed that nuclei extraction is the most efficient way to isolate cells from the periosteum and the fracture callus.

(3) reduction of isolation time and cell stress. Previous studies showed that enzymatic digestion causes cell stress and induces stem cell activation (Machado et al. 2021; van den Brink et al. 2017). Therefore, we decided to perform snRNAseq to analyze the transcriptome of the intact periosteum without digestion induced-biais.

We added this sentence in the result section: “Single nuclei transcriptomics was shown to provide results equivalent to single cell transcriptomics, but with better cell type representation and reduced digestion-induced stress response (Selewa et al. 2020; Wen et al. 2022; Ding et al. 2020; H. Wu et al. 2019; Machado et al. 2021)”.

The list of genes used for cell type mapping are presented in Figure 3 – Supplementary figure 1. We added a detailed dot plot as Figure 3 – Supplementary figure 2.

(2) During the fracture healing process of long bones, the influx of fibroblasts is a relatively common occurrence, and the fibrous callus that forms during bone repair and regeneration is reported to disappear over time. Therefore, inferring that IIFC differentiates into osteo- and chondrogenic cells based solely on their simultaneous appearance in the same time and space is challenging. More detailed validation is necessary, beyond what is supported by bioinformatics analysis.

The first step of bone repair is the formation of a fibrous callus, before cartilage and bone formation. There are no data in the literature demonstrating that an influx of fibroblasts occurs at the fracture site. Several studies now show that cells involved in callus formation are recruited locally (i.e. from the bone marrow, the periosteum and the skeletal muscle surrounding the fracture site) (Duchamp de Lageneste et al. 2018; Julien et al. 2021; Colnot 2009; Jeffery et al. 2022; Debnath et al. 2018; Matsushita et al. 2020; Julien et al. 2022; Matthews et al. 2021). The contribution of locally activated SSPCs to the fibrous callus is less well understood. Lineage tracing shows that GFP+ cell populations traced in Prx1Cre-GFP mice include SSPCs, IIFCs, chondrocytes and osteoblasts.

The timing of the cell trajectories observed in our dataset correlates with the timing of callus formation previously described in the literature as the day 3 post-fracture mostly contains IIFCs while chondrocytes and osteoblasts appear from day 5 post-fracture. We conclude that IIFCs differentiate into osteochondrogenic cells based on multiple evidence beside the simultaneous appearance in time and space:

In silico trajectory analyses identify a trajectory from SSPCs to osteochondrogenic cells via IIFCs. We added an analysis to show that our pseudotime trajectory parallels the timepoints of the dataset, confirming that the differentiation trajectory follows the timing of cell differentiation (Figure 5D).

- We show that IIFCs start to express chondrogenic and osteogenic genes prior to engaging into chondrogenesis and osteogenesis. In addition, we detected activation of osteo- and chondrogenic specific transcription factors in IIFCs. This shows a differentiation continuum between SSPCs, IIFCS, and osteochondrogenic cells (Figures 6-8).

- Using transplantation assay, we showed that IIFCs form cartilage and bone, therefore reinforcing the osteochondrogenic potential of this population (Figure 6B).

- IIFCs do not undergo apoptosis. We assessed the expression of apoptosis-related genes by IIFCs and did not detect expression. This was confirmed by cleaved caspase 3 immunostaining showing that a very low percentage of cells in the early fibrotic tissue undergo apoptosis.

Therefore, the idea that the initial fibrous callus is replaced by a new influx of SSPCs or committed progenitors is not supported by recent literature and is not observed in our dataset containing all cell types from the periosteum and fracture site. Overall, our bioinformatic analyses combined with our in vivo validation strongly support that IIFCs are differentiating into chondrocytes and osteoblasts during bone repair. Additional in vivo functional studies will aim to further validate the trajectory and investigate the critical factors regulating this process.

(3) The influx of most osteogenic progenitors to the bone fracture site typically appears after postfracture day 7. It's essential to ascertain whether the osteogenic cells observed at the time of this study differentiated from IIFC or migrated from surrounding mesenchymal stem cells.

As mentioned above, there is not clear evidence in the literature indicating an influx of osteoprogenitors. Cells involved in callus formation are recruited locally and predominantly from the periosteum (Duchamp de Lageneste et al. 2018; Julien et al. 2021; Colnot 2009; Jeffery et al. 2022; Debnath et al. 2018; Matsushita et al. 2020; Matthews et al. 2021; Julien et al. 2022). Our datasets therefore include all cell populations that form the callus. Other sources of SSPCs include the surrounding muscle that contributes mostly to cartilage, and bone marrow that contributes to a low percentage of the callus osteoblasts in the medullary cavity (Julien et al. 2021; Jeffery et al. 2022). We provide evidence that IIFCs give rise to osteogenic cells using our bioinformatic analyses and in vivo transplantation assay (listed in the response above). As indicated in our response to reviewer #1, the steps leading to osteogenic differentiation observed in our dataset reflect the first step of callus ossification and correspond to the process of intramembranous ossification (up to day 7 post-injury). Endochondral ossification also contributes to osteoblasts including the transdifferentiation of chondrocytes into osteoblasts (Julien et al. 2020; Zhou et al. 2014; Hu et al. 2017). While this process mostly occurs around day 14 postfracture, we begin to detect this transition in our integrated day 5-day 7 dataset as shown in Author response image 1.

(4) It's crucial to determine whether the IIFC appearing at the fracture site contributes to the formation of the callus matrix or undergoes apoptosis during the fracture healing process. In the early steps of bone repair, the callus is mostly composed of an extracellular matrix (ECM). IIFCs are expressing high levels of ECM genes, including *Postn*, *Aspn* and collagens (*Col3a1, Col5a1, Col8a1, Col12a1*) (Figure 3 – Supplementary Figures 1-2 and Fig. 7 – Supplementary Figure 1B). IIFCs are the cells expressing the highest levels of matrix-related genes compared to the other cell types in the fracture environment (i.e. immune cells, endothelial cells, Schwann cells, pericytes, …) as shown now in Fig. 7 – Supplementary Figure 1A. Therefore, IIFCs are the main contributors to the callus matrix.

We investigated if IIFCs undergo apoptosis. We observed that only a low percentage of IIFCs express apoptosis-related genes and are positive for cleaved caspase 3 immunostaining at days 3, 5 and 7 of bone repair. This shows that IIFCs do not undergo apoptosis and reinforces our model in which IIFCs further differentiate into osteoblasts and chondrocytes. We added these data in Fig. 7 – Supplementary Figure 2 and added the sentence in the results section “Only a small subset of IIFCs undergo apoptosis, further supporting that IIFCs are maintained in the fracture environment giving rise to osteoblasts and chondrocytes (Fig. 7 – Supplementary Figure 2).”

(5) Results from the snRNA seq highlight the paracrine role of IIFC, and verification is needed to ensure that the effect this has on surrounding osteogenic lineages is not misinterpreted.

To assess cell-cell interactions, we used tools such as Connectome and CellChat to infer and quantify intercellular communication networks between cell types. Studies showed the robustness of these tools combined with in vivo validation (Sinha et al. 2022; Alečković et al. 2022; Li et al. 2023). Here we used these tools to illustrate the paracrine profile of IIFCs, but in vivo validation would be required using gene inactivation to assess the requirement of individual paracrine factors. We performed extensive analyses of the crosstalk between immune cells and SSPCs using our dataset in another study combined with in vivo validation, showing the robustness of the tool and the dataset (Hachemi et al. 2024). We adjusted our conclusions to reflect our analyses: “suggesting a crucial paracrine role of this transient IIFC population during fracture healing”, “suggesting their central role in mediating cell interactions after fracture”, “suggesting that SSPCs can receive signals from IIFC”.

References

Aghajanian, Patrick, et Subburaman Mohan. 2018. “The Art of Building Bone: Emerging Role of Chondrocyte-to-Osteoblast Transdifferentiation in Endochondral Ossification“. *Bone Research* 6 (1): 19. https://doi.org/10.1038/s41413-018-0021-z.

Alečković, Maša, Simona Cristea, Carlos R. Gil Del Alcazar, Pengze Yan, Lina Ding, Ethan D. Krop, Nicholas W. Harper, et al. 2022. “Breast Cancer Prevention by Short-Term Inhibition of TGFβ Signaling“. *Nature Communications* 13 (1): 7558. https://doi.org/10.1038/s41467-02235043-5.

Ambrosi, Thomas H., Owen Marecic, Adrian McArdle, Rahul Sinha, Gunsagar S. Gulati, Xinming Tong, Yuting Wang, et al. 2021. “Aged Skeletal Stem Cells Generate an Inflammatory Degenerative Niche”. *Nature* 597 (7875): 256‑62. https://doi.org/10.1038/s41586-021-03795-7.

Baccin, Chiara, Jude Al-Sabah, Lars Velten, Patrick M. Helbling, Florian Grünschläger, Pablo Hernández-Malmierca, César Nombela-Arrieta, Lars M. Steinmetz, Andreas Trumpp, et Simon Haas. 2020. “Combined Single-Cell and Spatial Transcriptomics Reveal the Molecular, Cellular and Spatial Bone Marrow Niche Organization”. *Nature Cell Biology* 22 (1): 38‑48. https://doi.org/10.1038/s41556-019-0439-6.

Balemans, Wendy, et Wim Van Hul. 2007. “The Genetics of Low-Density Lipoprotein ReceptorRelated Protein 5 in Bone: A Story of Extremes”. *Endocrinology* 148 (6): 2622‑29. https://doi.org/10.1210/en.2006-1352.

Brink, Susanne C van den, Fanny Sage, Ábel Vértesy, Bastiaan Spanjaard, Josi Peterson-Maduro, Chloé S Baron, Catherine Robin, et Alexander van Oudenaarden. 2017. “Single-Cell Sequencing Reveals Dissociation-Induced Gene Expression in Tissue Subpopulations”. *Nature Methods* 14 (10): 935‑36. https://doi.org/10.1038/nmeth.4437.

Cao, Junjie, Yalin Wei, Jing Lian, Lunyun Yang, Xiaoyan Zhang, Jiaying Xie, Qiang Liu, Jinyong Luo, Baicheng He, et Min Tang. 2017. ”Notch Signaling Pathway Promotes Osteogenic Differentiation of Mesenchymal Stem Cells by Enhancing BMP9/Smad Signaling”. *International Journal of Molecular Medicine* 40 (2): 378‑88. https://doi.org/10.3892/ijmm.2017.3037.

Cao, Junyue, Malte Spielmann, Xiaojie Qiu, Xingfan Huang, Daniel M. Ibrahim, Andrew J. Hill, Fan Zhang, et al. 2019. ”The Single-Cell Transcriptional Landscape of Mammalian Organogenesis”. *Nature* 566 (7745): 496‑502. https://doi.org/10.1038/s41586-019-0969-x.

Colnot, Céline. 2009. “Skeletal Cell Fate Decisions Within Periosteum and Bone Marrow During Bone Regeneration”. *Journal of Bone and Mineral Research* 24 (2): 274‑82. https://doi.org/10.1359/jbmr.081003.

Debnath, Shawon, Alisha R. Yallowitz, Jason McCormick, Sarfaraz Lalani, Tuo Zhang, Ren Xu, Na Li, et al. 2018. “Discovery of a Periosteal Stem Cell Mediating Intramembranous Bone Formation”. *Nature* 562 (7725): 133‑39. https://doi.org/10.1038/s41586-018-0554-8.

Ding, Jiarui, Xian Adiconis, Sean K. Simmons, Monika S. Kowalczyk, Cynthia C. Hession, Nemanja D. Marjanovic, Travis K. Hughes, et al. 2020. “Systematic Comparison of Single-Cell and Single-Nucleus RNA-Sequencing Methods”. *Nature Biotechnology* 38 (6): 737‑46.

https://doi.org/10.1038/s41587-020-0465-8.

Dishowitz, Michael I., Shawn P. Terkhorn, Sandra A. Bostic, et Kurt D. Hankenson. 2012. “Notch Signaling Components Are Upregulated during Both Endochondral and Intramembranous Bone Regeneration”. *Journal of Orthopaedic Research* 30 (2): 296‑303. https://doi.org/10.1002/jor.21518.

Duchamp de Lageneste, Oriane, Anaïs Julien, Rana Abou-Khalil, Giulia Frangi, Caroline Carvalho, Nicolas Cagnard, Corinne Cordier, Simon J. Conway, et Céline Colnot. 2018. “Periosteum Contains Skeletal Stem Cells with High Bone Regenerative Potential Controlled by Periostin”. *Nature Communications* 9 (1): 773. https://doi.org/10.1038/s41467-018-03124-z.

Hsieh, Chen-Chan, B. Linju Yen, Chia-Chi Chang, Pei-Ju Hsu, Yu-Wei Lee, Men-Luh Yen, ShawFang Yet, et Linyi Chen. 2023. “Wnt Antagonism without TGFβ Induces Rapid MSC Chondrogenesis via Increasing AJ Interactions and Restricting Lineage Commitment”. *iScience* 26 (1): 105713. https://doi.org/10.1016/j.isci.2022.105713.

Hu, Diane P., Federico Ferro, Frank Yang, Aaron J. Taylor, Wenhan Chang, Theodore Miclau, Ralph S. Marcucio, et Chelsea S. Bahney. 2017. “Cartilage to Bone Transformation during Fracture Healing Is Coordinated by the Invading Vasculature and Induction of the Core Pluripotency Genes”. *Development* 144 (2): 221‑34. https://doi.org/10.1242/dev.130807.

Jeffery, Elise C., Terry L.A. Mann, Jade A. Pool, Zhiyu Zhao, et Sean J. Morrison. 2022. “Bone Marrow and Periosteal Skeletal Stem/Progenitor Cells Make Distinct Contributions to Bone Maintenance and Repair”. *Cell Stem Cell* 29 (11): 1547-1561.e6. https://doi.org/10.1016/j.stem.2022.10.002.

Julien, Anais, Anuya Kanagalingam, Ester Martínez-Sarrà, Jérome Megret, Marine Luka, Mickaël Ménager, Frédéric Relaix, et Céline Colnot. 2021. “Direct contribution of skeletal muscle mesenchymal progenitors to bone repair”. *Nature Communications* 12 (1): 2860. https://doi.org/10.1038/s41467-021-22842-5.

Julien, Anais, Simon Perrin, Oriane Duchamp de Lageneste, Caroline Carvalho, Morad Bensidhoum, Laurence Legeai-Mallet, et Céline Colnot. 2020. “FGFR3 in Periosteal Cells Drives Cartilage-to-Bone Transformation in Bone Repair”. *Stem Cell Reports* 15 (4): 955‑67. https://doi.org/10.1016/j.stemcr.2020.08.005.

Julien, Anais, Simon Perrin, Ester Martínez-Sarrà, Anuya Kanagalingam, Caroline Carvalho, Marine Luka, Mickaël Ménager, et Céline Colnot. 2022. “Skeletal Stem/Progenitor Cells in Periosteum and Skeletal Muscle Share a Common Molecular Response to Bone Injury”. *Journal of Bone and Mineral Research*, juin, jbmr.4616. https://doi.org/10.1002/jbmr.4616.

Kang, Sona, Christina N. Bennett, Isabelle Gerin, Lauren A. Rapp, Kurt D. Hankenson, et Ormond A. MacDougald. 2007. “Wnt Signaling Stimulates Osteoblastogenesis of Mesenchymal Precursors by Suppressing CCAAT/Enhancer-Binding Protein α and Peroxisome Proliferator Activated Receptor γ”. *Journal of Biological Chemistry* 282 (19): 14515‑24. https://doi.org/10.1074/jbc.M700030200.

Komatsu, David E., Michelle N. Mary, Robert Jason Schroeder, Alex G. Robling, Charles H. Turner, et Stuart J. Warden. 2010. “Modulation of Wnt Signaling Influences Fracture Repair”. *Journal of Orthopaedic Research* 28 (7): 928‑36. https://doi.org/10.1002/jor.21078.

Hachemi, Yasmine, Simon Perrin, Maria Ethel, Anais Julien, Julia Vettese, Blandine Geisler, Christian Göritz, et Céline Colnot. 2024. “Multimodal Analyses of Immune Cells during Bone Repair Identify Macrophages as a Therapeutic Target in Musculoskeletal Trauma”. https://doi.org/10.1101/2024.04.29.591608.

Kraus, Jessica M., Dion Giovannone, Renata Rydzik, Jeremy L. Balsbaugh, Isaac L. Moss, Jennifer L. Schwedler, Julien Y. Bertrand, et al. 2022. “Notch Signaling Enhances Bone Regeneration in the Zebrafish Mandible”. *Development* 149 (5): dev199995. https://doi.org/10.1242/dev.199995.

Lee, S., L. H. Remark, A. M. Josephson, K. Leclerc, E. Muiños Lopez, D. J. Kirby, Devan Mehta, et al. 2021. “Notch-Wnt Signal Crosstalk Regulates Proliferation and Differentiation of Osteoprogenitor Cells during Intramembranous Bone Healing”. *Npj Regenerative Medicine* 6 (1): 29. https://doi.org/10.1038/s41536-021-00139-x.

Li, Jiaoduan, Dongyan Cao, Lixin Jiang, Yiwen Zheng, Siyuan Shao, Ai Zhuang, et Dongxi Xiang. 2023. “ITGB2-ICAM1 Axis Promotes Liver Metastasis in BAP1-Mutated Uveal Melanoma with Retained Hypoxia and ECM Signatures”. *Cellular Oncology (Dordrecht)*, décembre. https://doi.org/10.1007/s13402-023-00908-4.

Logan, Malcolm, James F. Martin, Andras Nagy, Corrinne Lobe, Eric N. Olson, et Clifford J. Tabin. 2002. “Expression of Cre Recombinase in the Developing Mouse Limb Bud Driven by aPrxl Enhancer”. *Genesis* 33 (2): 77‑80. https://doi.org/10.1002/gene.10092.

Machado, Léo, Perla Geara, Jordi Camps, Matthieu Dos Santos, Fatima Teixeira-Clerc, Jens Van Herck, Hugo Varet, et al. 2021.”Tissue Damage Induces a Conserved Stress Response That Initiates Quiescent Muscle Stem Cell Activation”. *Cell Stem Cell* 28 (6): 1125-1135.e7. https://doi.org/10.1016/j.stem.2021.01.017.

Matsushita, Yuki, Mizuki Nagata, Kenneth M. Kozloff, Joshua D. Welch, Koji Mizuhashi, Nicha Tokavanich, Shawn A. Hallett, et al. 2020. “A Wnt-Mediated Transformation of the Bone Marrow Stromal Cell Identity Orchestrates Skeletal Regeneration”. *Nature Communications* 11 (1): 332. https://doi.org/10.1038/s41467-019-14029-w.

Matthews, Brya G, Danka Grcevic, Liping Wang, Yusuke Hagiwara, Hrvoje Roguljic, Pujan Joshi, Dong-Guk Shin, Douglas J Adams, et Ivo Kalajzic. 2014. “Analysis of αSMA-Labeled Progenitor Cell Commitment Identifies Notch Signaling as an Important Pathway in Fracture Healing”. *Journal of Bone and Mineral Research* 29 (5): 1283‑94. https://doi.org/10.1002/jbmr.2140.

Matthews, Brya G, Sanja Novak, Francesca V Sbrana, Jessica L Funnell, Ye Cao, Emma J Buckels, Danka Grcevic, et Ivo Kalajzic. 2021. “Heterogeneity of Murine Periosteum Progenitors Involved in Fracture Healing”. *eLife* 10 (février):e58534. https://doi.org/10.7554/eLife.58534.

Minear, Steve, Philipp Leucht, Samara Miller, et Jill A Helms. 2010. “rBMP Represses Wnt Signaling and Influences Skeletal Progenitor Cell Fate Specification during Bone Repair”. *Journal of Bone and Mineral Research* 25 (6): 1196‑1207. https://doi.org/10.1002/jbmr.29.

Minear, Steven, Philipp Leucht, Jie Jiang, Bo Liu, Arial Zeng, Christophe Fuerer, Roel Nusse, et Jill A. Helms. 2010. “Wnt Proteins Promote Bone Regeneration”. *Science Translational Medicine* 2 (29). https://doi.org/10.1126/scitranslmed.3000231.

Novak, Sanja, Emilie Roeder, Benjamin P. Sinder, Douglas J. Adams, Chris W. Siebel, Danka Grcevic, Kurt D. Hankenson, Brya G. Matthews, et Ivo Kalajzic. 2020. “Modulation of Notch1 Signaling Regulates Bone Fracture Healing”. *Journal of Orthopaedic Research* 38 (11): 2350‑61. https://doi.org/10.1002/jor.24650.

Pinzone, Joseph J., Brett M. Hall, Nanda K. Thudi, Martin Vonau, Ya-Wei Qiang, Thomas J. Rosol, et John D. Shaughnessy. 2009. “The Role of Dickkopf-1 in Bone Development, Homeostasis, and Disease”. *Blood* 113 (3): 517‑25. https://doi.org/10.1182/blood-2008-03-145169.

Remark, Lindsey H., Kevin Leclerc, Malissa Ramsukh, Ziyan Lin, Sooyeon Lee, Backialakshmi Dharmalingam, Lauren Gillinov, et al. 2023. “Loss of Notch Signaling in Skeletal Stem Cells Enhances Bone Formation with Aging”. *Bone Research* 11 (1): 50. https://doi.org/10.1038/s41413-023-00283-8.

Ruscitto, Angela, Peng Chen, Ikue Tosa, Ziyi Wang, Gan Zhou, Ingrid Safina, Ran Wei, et al. 2023. “Lgr5-Expressing Secretory Cells Form a Wnt Inhibitory Niche in Cartilage Critical for Chondrocyte Identity”. *Cell Stem Cell* 30 (9): 1179-1198.e7. https://doi.org/10.1016/j.stem.2023.08.004.

Selewa, Alan, Ryan Dohn, Heather Eckart, Stephanie Lozano, Bingqing Xie, Eric Gauchat, Reem Elorbany, et al. 2020. “Systematic Comparison of High-Throughput Single-Cell and SingleNucleus Transcriptomes during Cardiomyocyte Differentiation”. *Scientific Reports* 10 (1): 1535. https://doi.org/10.1038/s41598-020-58327-6.

Sinha, Sarthak, Holly D. Sparks, Elodie Labit, Hayley N. Robbins, Kevin Gowing, Arzina Jaffer, Eren Kutluberk, et al. 2022. “Fibroblast Inflammatory Priming Determines Regenerative versus Fibrotic Skin Repair in Reindeer”. *Cell* 185 (25): 4717-4736.e25. https://doi.org/10.1016/j.cell.2022.11.004.

Wang, Cuicui, Jason A. Inzana, Anthony J. Mirando, Yinshi Ren, Zhaoyang Liu, Jie Shen, Regis J. O’Keefe, Hani A. Awad, et Matthew J. Hilton. 2016. “NOTCH Signaling in Skeletal Progenitors Is Critical for Fracture Repair”. *The Journal of Clinical Investigation* 126 (4): 1471‑81. https://doi.org/10.1172/JCI80672.

Wen, Fei, Xiaojie Tang, Lin Xu, et Haixia Qu. 2022. “Comparison of Single‑nucleus and Single‑cell Transcriptomes in Hepatocellular Carcinoma Tissue”. *Molecular Medicine Reports* 26 (5): 339. https://doi.org/10.3892/mmr.2022.12855.

Wu, Chia-Lung, Amanda Dicks, Nancy Steward, Ruhang Tang, Dakota B. Katz, Yun-Rak Choi, et Farshid Guilak. 2021. “Single Cell Transcriptomic Analysis of Human Pluripotent Stem Cell Chondrogenesis”. *Nature Communications* 12 (1): 362. https://doi.org/10.1038/s41467-02020598-y.

Wu, Haojia, Yuhei Kirita, Erinn L. Donnelly, et Benjamin D. Humphreys. 2019. “Advantages of Single-Nucleus over Single-Cell RNA Sequencing of Adult Kidney: Rare Cell Types and Novel Cell States Revealed in Fibrosis”. *Journal of the American Society of Nephrology* 30 (1): 23‑32. https://doi.org/10.1681/ASN.2018090912.

Zhong, Leilei, Lutian Yao, Robert J. Tower, Yulong Wei, Zhen Miao, Jihwan Park, Rojesh Shrestha, et al. 2020. “Single Cell Transcriptomics Identifies a Unique Adipose Lineage Cell Population That Regulates Bone Marrow Environment”. *eLife* 9 (avril):e54695. https://doi.org/10.7554/eLife.54695.

Zhou, Xin, Klaus von der Mark, Stephen Henry, William Norton, Henry Adams, et Benoit de Crombrugghe. 2014. “Chondrocytes Transdifferentiate into Osteoblasts in Endochondral Bone during Development, Postnatal Growth and Fracture Healing in Mice”. Édité par Matthew L. Warman. *PLoS Genetics* 10 (12): e1004820. https://doi.org/10.1371/journal.pgen.1004820.